# The GPR171 pathway suppresses T cell activation and limits antitumor immunity

Yuki Fujiwara [1], Robert J. Torphy [1], Yi Sun[1], Emily N. Miller [1], Felix Ho[1], Nicholas Borcherding[2], Tuoqi Wu [3], Raul M. Torres[3], Weizhou Zhang [4], Richard D. Schulick[1] & Yuwen Zhu [1✉]

The recently identified G-protein-coupled receptor GPR171 and its ligand BigLEN are thought to regulate food uptake and anxiety. Though GPR171 is commonly used as a T cell signature gene in transcriptomic studies, its potential role in T cell immunity has not been explored. Here we show that GPR171 is transcribed in T cells and its protein expression is induced upon antigen stimulation. The neuropeptide ligand BigLEN interacts with GPR171 to suppress T cell receptor-mediated signalling pathways and to inhibit T cell proliferation. Loss of GPR171 in T cells leads to hyperactivity to antigen stimulation and GPR171 knockout mice exhibit enhanced antitumor immunity. Blockade of GPR171 signalling by an antagonist promotes antitumor T cell immunity and improves immune checkpoint blockade therapies. Together, our study identifies the GPR171/BigLEN axis as a T cell checkpoint pathway that can be modulated for cancer immunotherapy.

---

[1] Department of Surgery, University of Colorado Anschutz Medical Campus, Aurora, CO 80045, USA. [2] Department of Pathology and Immunology, Washington University, St. Louis, MO 63110, USA. [3] Department of Immunology and Microbiology, University of Colorado Anschutz Medical Campus, Aurora, CO 80045, USA. [4] Department of Pathology, University of Florida, Gainesville, FL 32610, USA. ✉email: yuwen.zhu@cuanschutz.edu

mmune checkpoints, also known as coinhibitory molecules, are a group of cell-surface molecules that control T-cell responses[1,2]. The majority of known immune checkpoints are members of the immunoglobulin superfamily (IgSF), which include at least CTLA-4, PD-1, LAG3, VISTA, TIGIT, and CD112R[3–6]. Multiple checkpoint receptors are upregulated on tumor-infiltrating lymphocytes (TILs) and contribute to T-cell dysfunction in the tumor immune microenvironment (TIME)[7]. Immune checkpoint blockers (ICB), including antibodies against CTLA-4 and PD-1/PD-L1, have revolutionized cancer treatment[8–10]. However, overall response rate to ICB in many cancers remains low, as tumor cells often evade immune attack through multiple suppressive and tolerogenic strategies[11,12]. Thus, characterization of additional T-cell checkpoints and careful analysis of their individual roles in the TIME will likely have important therapeutic applications.

G-protein-coupled receptors (GPCRs) are one of the largest surface receptor superfamilies and many of them play important roles in immune responses, including chemokine receptors[13,14]. Several GPCR members are known to directly modulate T-cell receptor (TCR) signaling to regulate T-cell response, including adenosine A2A receptors (A2AR)[14,15], lysophosphatidic acid receptor 5 (LPA5)[16,17], and members of the P2Y receptors[18,19]. G-protein-coupled receptor 171 (GPR171) is a GPCR receptor with close homology to the P2Y receptors[20] and is reported to regulate feeding and anxiety behaviors[21,22]. BigLEN, a 16-amino acid neuropeptide derived from a neuroendocrine secretory pathway protein proSAAS, has been identified as the endogenous ligand for GPR171[21]. On the other hand, GPR171 is a well-accepted T-cell signature gene in cancer database analysis, though its potential immune function is untested[23–25]. Based on early published databases, *GPR171* expression tends to be upregulated in tumor-infiltrating lymphocytes (TILs) and its expression in melanoma patients is significantly increased in response to immunotherapies[26,27]. All these imply a possible role of GPR171 in T-cell immunity.

Here, we show that GPR171 expression is inducible in T cells, and T-cell- mediated immune responses are suppressed via GPR171 signaling. Furthermore, disruption of GPR171 signaling promotes T-cell-mediated antitumor immunity.

## Results

**Signaling through GPR171 inhibits human T-cell response**. We investigated whether GPR171 is a gene enriched in human T cells. The microarray data[28] derived from the BioGPS database revealed that human *GPR171* is preferentially expressed in CD8+ and CD4+ T-cell subsets (Supplementary Fig. 1a, http://biogps.org/gene/29909/). In line with that, *GPR171* expression is well associated with T-cell signature genes *CD3e* and *LCK* across multiple cancer types in The Cancer Genome Atlas (TCGA) database (Supplementary Fig. 1b). In a single-cell RNAseq analysis of immune cell infiltrates in human melanoma[29], *GPR171* was preferentially expressed in T cells and NK cells (Supplementary Fig. 1c). In patients with human liver cancer[30], GPR171 transcript was more abundant in tumor-infiltrating T cells than those in peripheral blood (Supplementary Fig. 1d); among different T-cell subsets, effector T cells and T cells with an exhausted phenotype exhibited higher levels of *GPR171* transcripts (Supplementary Fig. 1e). We sorted out different immune cell subtypes from peripheral blood mononuclear cells (PBMCs) of a healthy donor to examine GPR171 expression; *GPR171* transcript was abundant in both T-cell subsets and NK cells, while it was undetectable in B cells or CD14+ monocytes (Fig. 1a and Supplementary Fig. 2a). Consistently, our western blot result indicated that Jurkat cells, a human T-cell leukemia line, and normal PBMCs both expressed significant amount of GPR171 protein (Fig. 1b).

We hypothesized that GPR171 is an understudied functional receptor for T cells. To test that, we utilized neuropeptide BigLEN, a known natural ligand for GPR171[21], to amplify GPR171 signal in a human T-cell proliferation assay. Under the stimulation of a human CD3 mAb (clone OKT3), the presence of BigLEN in human CD8+ T-cell culture led to fewer divided T cells (Fig. 1c), accompanied with a smaller number of recovered live cells upon 6 days stimulation (Fig. 1d). The proliferation of CD4+ T cells under the stimulation of different concentrations of OKT3 was constantly inhibited by BigLEN (Fig. 1e); as a result, in the presence of BigLEN, human CD4+ T cells produced significantly less cytokines, including IL-2, IL-5, IL-9, IL-10, IL-17F, IL-21, TNF-α, and IFN-γ (Fig. 1f).

We further investigated the signaling pathways BigLEN possibly targets to inhibit T-cell response. BigLEN was able to inhibit OKT3- stimulated calcium flux in human T cells (Supplementary Fig. 2b). To validate the involvement of GPR171 in BigLEN- suppressed T-cell response, a GPR171 knockout (GPR171^KO) Jurkat cell line was generated via the CRISPR/Cas9 technology, and the loss of GPR171 expression was verified by western blot (Supplementary Fig. 2c). BigLEN suppressed TCR- triggered calcium flux in wild type (WT) Jurkat cells in a dose-dependent fashion; however, the suppressive effect of BigLEN on calcium flux was abolished in GPR171^KO Jurkat cells (Fig. 1g). The increase of intracellular calcium in T cells directly leads to downstream NFAT activation[31]. We then evaluated the NFAT activity by co-culturing CHO stimulator cells[32] expressing human PD-L1 (CHO/PD-L1) with Jurkat reporter cells, which was stably transduced with PD-1 and a luciferase reporter under the control of a NFAT response element. Blockade of the PD-L1 signal by a human PD-1 mAb increased NFAT activity in both control and GPR171 ^KO Jurkat cells under the stimulation of CHO/PD-L1 cells; however, the presence of BigLEN selectively reduced NFAT activity in control Jurkat cells, but not GPR171^KO cells (Fig. 1h). Consistently, BigLEN inhibited OKT3-stimulated phosphorylation of PLCγ1, ERK, ZAP70, CD3, and to a lesser extent AKT in control Jurkat cells, but not in GPR171^KO Jurkat cells (Fig. 1i, j). We were able to confirm the suppressive effect of BigLEN on TCR-stimulated signaling pathways in primary human T cells (Supplementary Fig. 2d, e). Altogether, our results suggest that BigLEN inhibits TCR-triggered signaling through human GPR171.

**GPR171 mediates the suppressive effect of BigLEN on T cell**. The expression of mouse GPR171 in immune cells seems to be not limited to T and NK cells[28] (Supplementary Fig. 3a). Our PCR analysis of sorted immune cell subsets from unstimulated mouse splenocytes supported that mouse T cells and NK cells expressed GPR171; in contrast to human B cells, mouse splenic B cells had abundant levels of GPR171 transcripts (Supplementary Fig. 3b, c). We confirmed a suppressive role for mouse BigLEN on T-cell response. When CFSE-labeled OT-1 T cells were stimulated by OVA257-264 peptide (SIINFEKL) in vitro, BigLEN was capable of inhibiting T-cell division in a dose-dependent manner (Fig. 2a); in parallel, the stimulation of OT-1 cells by different concentrations of SIINFEKL was significantly suppressed by BigLEN when we enumerated the number of live OT-1 T cells (Fig. 2b). Similarly, BigLEN inhibited the division and expansion of conventional CD4+ T cells from naive B6 mice in response to titrated CD3 mAb (Supplementary Fig. 3d, e). When we measured cytokines produced by mouse CD4+ T cells, the presence of BigLEN significantly reduced the majority of T-cell-produced cytokines, including IL-2, IFN-γ, TNF-α, IL-10, IL-17, and IL-22 (Supplementary Fig. 3f).

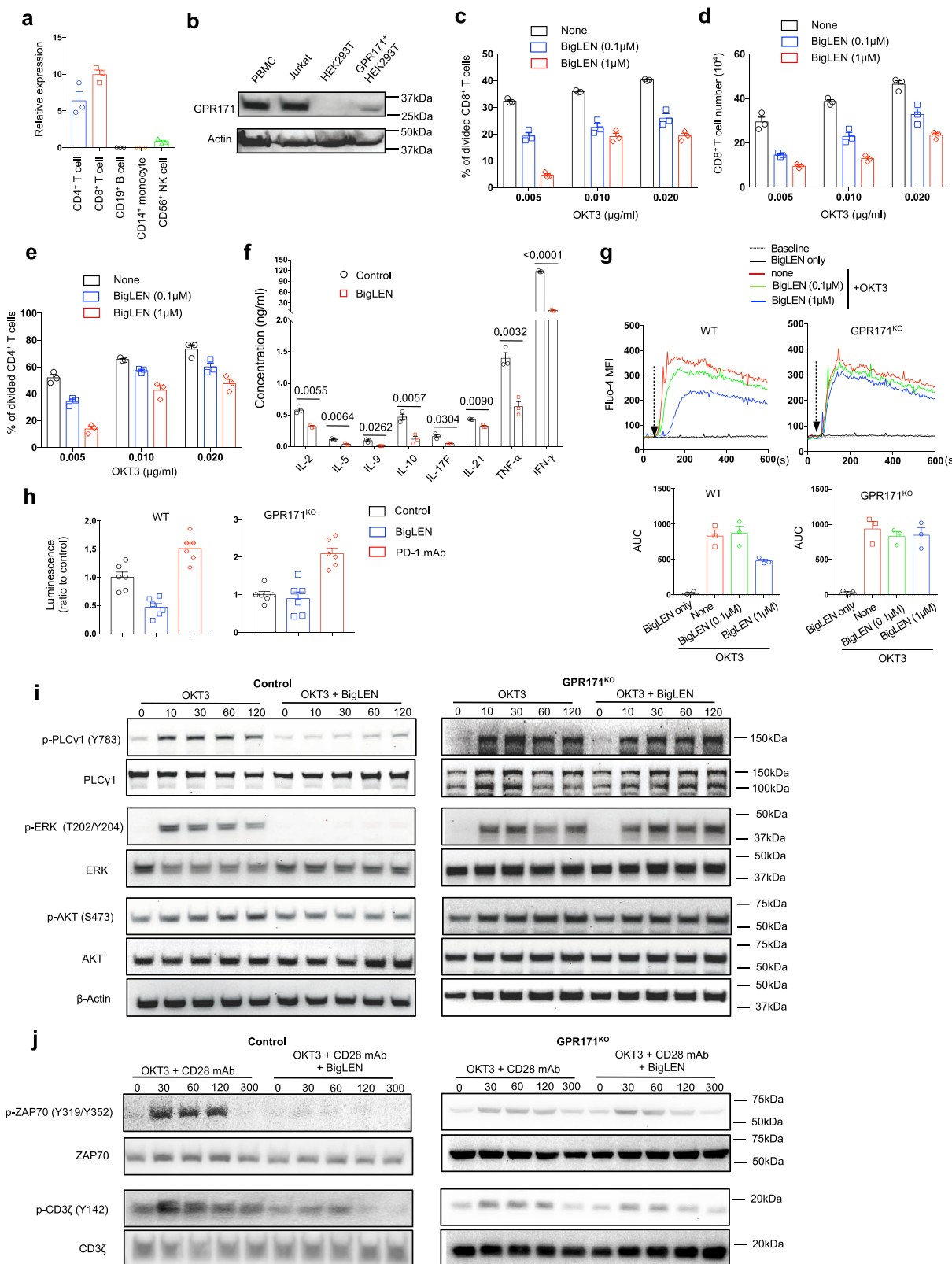

To further investigate the effect of the BigLEN/GPR171 pathway on T-cell-mediated immune response in vivo, we utilized a GPR171 antagonist MS0021570 that has been used to block the BigLEN and GPR171 interaction in a food uptake model[22]. Because of the essential role of immune checkpoints in T-cell tolerance[2], we hypothesized that GPR171 signaling is indispensable for the induction of T-cell anergy. In a peptide-induced T-cell anergy model[33], we transferred CD45.1+ OT-1 T cells into B6 mice and intravenously challenged with OVA257-264 peptide. Peptide administration alone led to a dramatic expansion of transferred OT-1 T cells on day 4 and then a quick contraction on day 7; more transferred OT-1 T cells were constantly detected in peripheral blood when GPR171 antagonist was administered at the beginning of peptide inoculation (Fig. 2c).

**Fig. 1 GPR171 signaling inhibits human T-cell activation. a** GPR171 transcript in different immune cells isolated from peripheral blood of healthy donor was determined by qPCR. **b** GPR171 protein in human PBMC and Jurkat cells were examined by western blot. HEK293T cells transduced to express GPR171(GPR171 + HEK293T) were used as a positive control. **c–f** Human T cells were stimulated with coated OKT3 with or without BigLEN. The percentage of divided CD8+ T cells (**c**) and the number of live cells (**d**), were determined 6 days after stimulation. 6 days post-stimulation, CD4+ T-cell division in response to titrated OKT3 with or without BigLEN was determined (**e**). Cytokines in the supernatant were quantified (**f**). Except IL-2 (day 2), all other cytokines were measured in CD4+ T-cell supernatant after 6 days culture. **g** Intracellular calcium levels over time in WT or GPR171KO Jurkat cells stimulated with OKT3 and different concentration of BigLEN (upper). The area under the curve (AUC) from the calcium flux test was calculated (lower). The dashed arrow indicated the addition of OKT3. **h** Luciferase activity of WT or GPR171KO Jurkat-NFAT-Luc cells stimulated by PD-L1+ CHO stimulator for 4 h. BigLEN or PD-1 mAb was added at the beginning of culture. **i** Western blot of phosphorylated PLCγ1, ERK and AKT in WT or GPR171KO Jurkat cells stimulated with OKT3 or OKT3 + BigLEN at different timepoints. **j** Western blot of phosphorylated ZAP70 and CD3ζ in WT or GPR171KO Jurkat cells stimulated with OKT3 + CD28 mAbs or OKT3 + CD28mAbs + BigLEN at different timepoints. Statistical significance was determined by two-tailed Student's *t*-test for **f**. Unless otherwise denoted, values are mean ± SEM. Source data was provided as a Source Data file. Data are representative of two (**a**, **b**, **c**, **d**, **e**, **f**, **g**, **h**, and **j**) or three independent experiments (**i**).

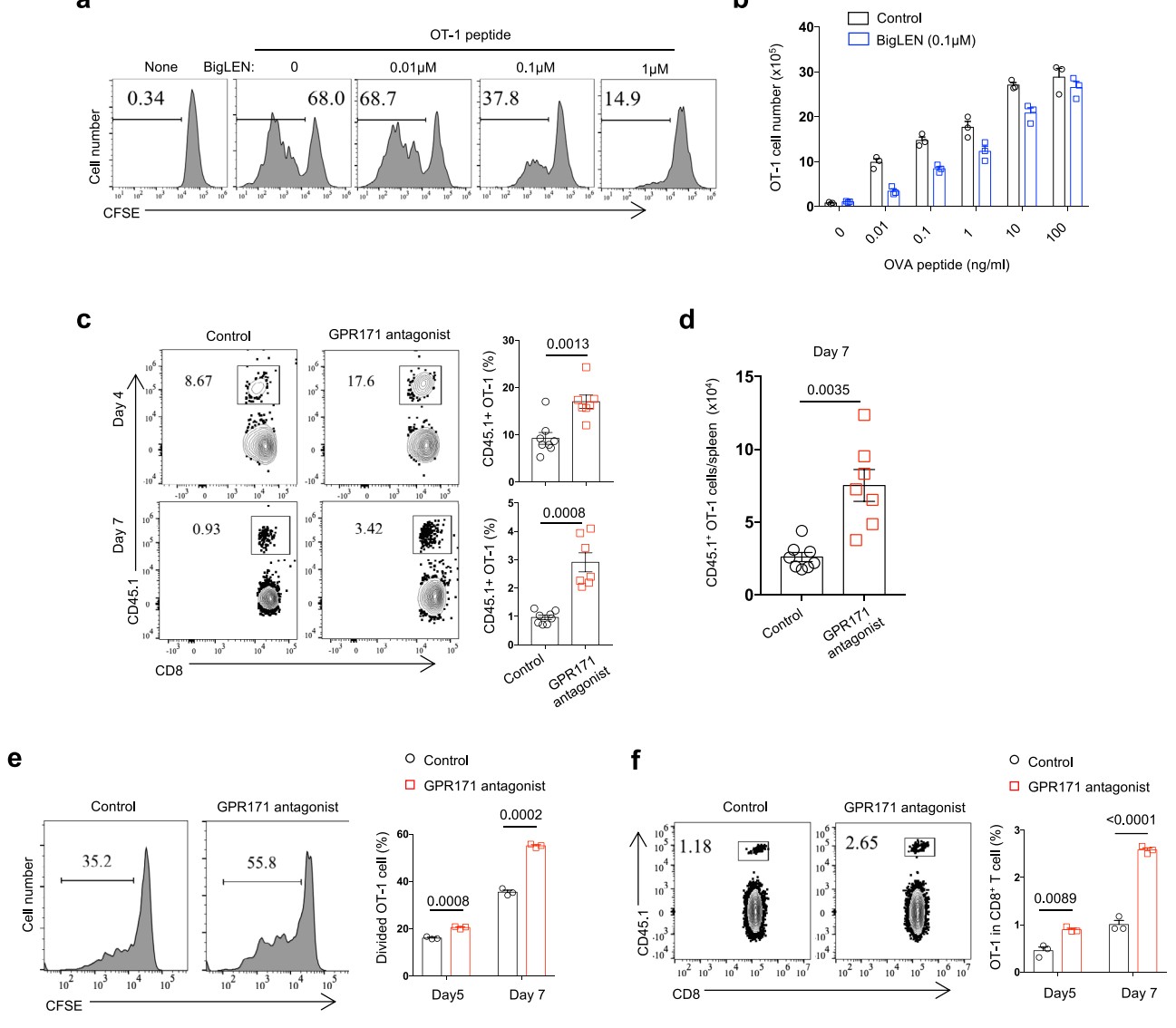

**Fig. 2 The BigLEN/GPR171 interaction inhibits mouse T-cell response. a**, **b** CFSE-labeled naive OT-1 T cells were stimulated with different concentrations of SIINFEKL peptide for 6 days. The percentages of divided OT-1 cells (**a**) and the numbers of live OT-1 cells (**b**) were determined by flow cytometry. **c**, **d** B6 mice transferred with naive CD45.1+ OT-1 T cells were immunized with OT-1 peptide and followed with the treatment of GPR171 antagonist. Transferred OT-1 T cells in peripheral blood at 4 and 7 days after peptide injection were examined by flow cytometry (**c**). The number of CD45.1+ OT-1 T cells in the spleen at 7 days after injection was enumerated (**d**). **e**, **f** B16-OVA bearing mice were transferred with CFSE-labeled CD45.1+ naive OT-1 T cells and followed with the treatment of GPR171 antagonist. 5 and 7 days after OT-1 transfer, the divisions (**e**) and percentages (**f**) of transferred OT-1 T cells in dLNs were determined. **c**, **d** n = 8 (control) and n = 7 (GPR171 antagonist) biologically independent samples. **e**, **f** n = 3 biologically independent samples. Statistical significance was determined by two-tailed Student's *t*-test for **c**, **d**, **e**, **f**. Unless otherwise denoted, values are mean ± SEM. Source data was provided as a Source Data file. All data are representative of two independent experiments.

Under the treatment of GPR171 antagonist, there were more OT-1 T cells present in the spleen 7 days after peptide challenge (Fig. 2d). This implied an active role of GPR171 signaling in the induction of T-cell anergy. To examine the role of GPR171 on T-cell priming, we transferred CFSE-labeled CD45.1+ naive OT-1 T cells into B6 mice bearing palpable B16-OVA tumors and followed with the treatment of GPR171 antagonist. When we examined draining lymph nodes (dLNs) on days 5 and 7 after T-cell transfer, the treatment of GPR171 antagonist to block GPR171 signaling constantly increased the divisions of transferred OT-1 T cells (Fig. 2e). As a result, there were about twice the frequency of transferred CD45.1+ OT-1 cells in the dLNs of mice 7 days after GPR171 antagonist treatment (Fig. 2f), compared to those of control mice. Thus, blockade of GPR171 signaling with an antagonist increases antigen-specific T-cell response in vivo.

**T cells from GPR171-deficient mice are hyperactive to antigen stimulation.** We generated GPR171-lacZ knock-in mice to better examine GPR171 expression and function in the immune system. The knock-in lacZ replaced the whole protein coding region of GPR171 gene (Supplementary Fig. 4a), which allowed us to determine the expression of mouse GPR171 by X-gal staining or detecting β-galactosidase (β-gal) by flow cytometry. Founder heterozygous (GPR171[+/lacZ]) mice were selected and were then interbred to produce homozygous (GRP171[lacZ/LacZ]) offspring that were examined by genomic PCR (Supplementary Fig. 4b). We examined GPR171 expression in tissues of adult GPR171[+/lacZ] mice by X-gal staining. We detected little to no X-gal staining in the liver, kidney, or brain (Supplementary Fig. 4c), though GPR171 was reported to be expressed in the brain[21]. Thymus and small intestine were strongly positive for GPR171 expression while the spleen exhibited weak X-gal signal (Supplementary Fig. 4c, d). We further performed flow cytometry analysis of β-gal to examine GPR171 expression in immune cells using GPR171[+/lacZ] mice. The expression of GPR171 variated at different stages of T-cell development; in the GPR171[+/lacZ] thymus, single-positive (CD4+CD8− or CD4−CD8+) thymocytes expressed the highest level of GPR171, followed by double-positive (CD4+ CD8+) thymocytes while GPR171 was almost undetectable in the double-negative (CD4−CD8−) population (Fig. 3a); we were able to further confirm that by quantitative PCR (Supplementary Fig. 4e). Flow cytometry analysis of single-cell suspensions from small intestine revealed that GPR171 was broadly expressed in intraepithelial lymphocytes (IELs), including TCRγδ+, TCRαβ+ CD8αα, TCRαβ+ CD8αβ, and TCRαβ+ CD4+ T cells (Supplementary Fig. 4f, g). In the spleen of naive GPR171[+/lacZ] mice, GPR171 (β-gal) was barely detectable in immune cells, including T cells, B cells, NK cells, dendritic cells (DC), and monocytes (Supplementary Fig. 4h, i). Under the stimulation of coated CD3 mAb for 2 days, CD4+ and CD8+ T cells in both WT (GPR171[+/+]) and GPR171[+/LacZ] mice were fully activated to express PD-1, while only T cells from GPR171[+/LacZ] mice expressed significant levels of β-gal (GPR171) (Fig. 3b, c). At the same time, the induction of GPR171 on activated GPR171[+/LacZ] T cells was verified by X-gal staining (Supplementary Fig. 4j). Similarly, GPR171 expression on NK cells was inducible by activation, as we detected clear upregulation of GPR171 on splenic NK cells from mice injected with poly I:C overnight (Fig. 3d and Supplementary Fig. 4k). Therefore, mouse GPR171 is an inducible receptor on both T and NK cells.

GPR171[lacZ/lacZ] were born at the expected Mendelian ratio and had no defect in weight gain or food intake (Supplementary Fig. 4l). GPR171[lacZ/lacZ] mice displayed no overt abnormality in thymic T-cell development (Supplementary Fig. 4m); the immune cell composition in secondary lymph organs was similar between GPR171[lacZ/lacZ] and WT littermates (Supplementary Fig. 4n). When we stimulated T cells with titrated CD3 mAb, both CD4+ and CD8+ T cells from GPR171[lacZ/lacZ] mice proliferated more vigorously than those from WT littermates (Fig. 3e). As expected, the inclusion of BigLEN greatly inhibited the divisions of WT CD4+ and CD8+ T cells; however, T cells from GPR171[lacZ/lacZ] mice were unresponsive to BigLEN stimulation (Fig. 3f), supporting that GPR171 mediates the suppressive effect of BigLEN on T cells.

To directly assess the impact of GPR171 on T-cell immunity in vivo, we utilized T cells from GPR171[lacZ/lacZ] mice in C57BL/6 background as donor cells in an acute graft versus host (GVH) model. In this model, splenocytes from GPR171[lacZ/lacZ] mice or WT littermates were adoptively transferred into B6D2F1/J mice and expanded in response to host H-2[d] MHC antigens[34]. Donor CD4+ and CD8+ T cells from GPR171[lacZ/lacZ] mice expanded faster than WT T-cell counterparts, as we examined donor T cells (H-2[d]) in peripheral blood at different timepoints (Fig. 3g and Supplementary Fig. 4o). When we analyzed splenocytes 10 days post-transfer, there was about 2.5-fold increases of donor CD4+ and CD8+ T cells in spleens transferred with GPR171[lacZ/lacZ] splenocytes, compared to those with WT splenocytes (Fig. 3h). We assessed the alloreactive killing capacity of donor cells by incubating isolated splenocytes with CFSE-labeled BALB/c splenocytes for 4 h. As shown in Fig. 3i, splenocytes grafted with GPR171[lacZ/lacZ] cells displayed much stronger alloreactive killing activity against BALB/c target cells than control splenocytes transferred with WT donor cells. Taken together, our results support that loss of GPR171 signaling leads to overactive T-cell response in vivo.

**Loss of GPR171 leads to improved antitumor immunity.** The inhibitory functions of GPR171 on T-cell immunity prompted us to look at the antitumor effect in GPR171[lacZ/lacZ] mice. First, we detected β-gal to determine GPR171 expression in GPR171[lacZ/lacZ] mice, which had been inoculated with MC38 tumors for 2 weeks. In draining lymph nodes (dLNs), there was a small percentage of GPR171-positive T cells, which were presumably tumor-reactive T cells; over half of intratumoral T cells, including both CD4+ and CD8+ T-cell subsets, expressed GPR171 (Fig. 4a). Similarly, there were significantly more percentages of GPR171-positive NK cells in tumors than those in the dLN (Fig. 4a). We further examined GPR171 expression together with other known immune checkpoints in TILs of MC38 tumors in GPR171[lacZ/lacZ] mice; the majorities of MC38 TILs of GPR171[lacZ/lacZ] mice were PD-1 positive, however, the expressions of LAG3, TIGIT, and TIM3 were primarily present in GPR171 (β-gal)-positive cells in both CD4+ and CD8+ T-cell subsets (Fig. 4b). Thus, the enrichment of GPR171 expression in TILs implies an important role of GPR171 in tumor immunity.

We inoculated GPR171[lacZ/lacZ] and littermates with MC38 tumors subcutaneously to assess the effect of GPR171 signaling on tumor immunity. GPR171[+/lacZ] mice and WT mice displayed comparable tumor growth curves while tumor growth was significantly retarded in GPR171[lacZ/lacZ] mice (Fig. 4c). As a result, tumor weights in GPR171[lacZ/lacZ] mice after 16 days inoculation were less than 1/3 of those from WT or GPR171[+/LacZ] littermates (Fig. 4d). Flow cytometry analysis of tumor-infiltrating immune cells indicated that GPR171[lacZ/lacZ] mice triggered a stronger antitumor immunity within the tumor. CD3+ T cells, particularly CD8+ T cells, were significantly increased in tumors of GPR171[lacZ/lacZ] mice; except a slight increase of NK cells, we did not observe clear changes in other non-T immune cells in tumors of GPR171[lacZ/lacZ] mice (Fig. 4e and Supplementary

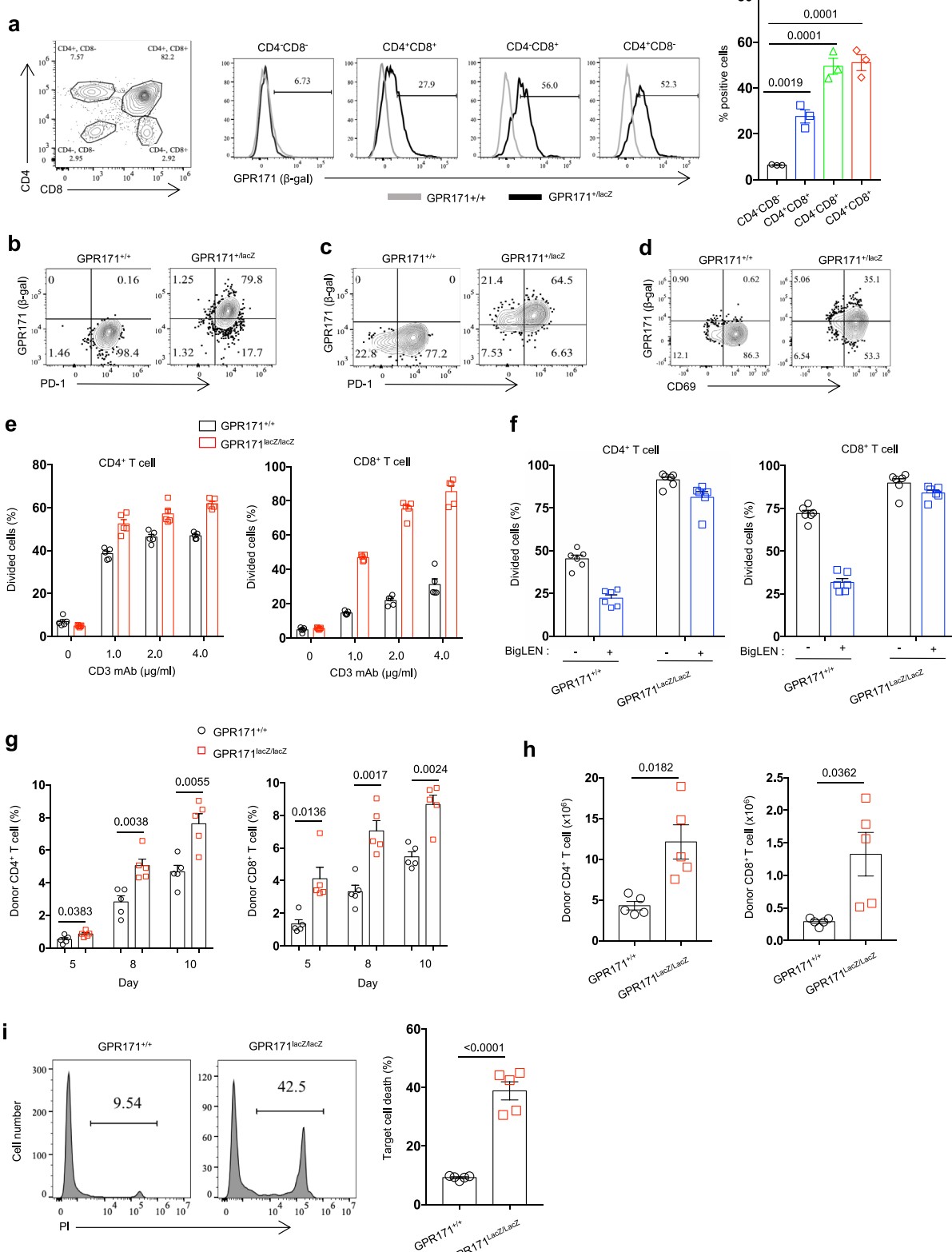

Fig. 5a, b). Though the density of intratumoral CD4+ T cells in *GPR171*lacZ/lacZ mice was not significantly altered, the proportion of FoxP3+ Treg cells in CD4+ T cells was markedly reduced (Fig. 4f); as a result, there was a significantly higher CD8+ T cells/ Treg ratio in MC38 tumors of *GPR171*lacZ/lacZ mice (Fig. 4g). CD8+ T cells in *GPR171*lacZ/lacZ tumors exhibited more effector T-cell functions, as they expressed proportionally more cytotoxic

molecule granzyme B and CD137 (Fig. 4h), a key costimulatory receptor whose expression is associated with effector CD8+ T-cell activation[35]. We depleted individual immune cell subset by mAb to assess their respective role in enhanced antitumor response in *GPR171*LacZ/LacZ mice. As shown in Fig. 4i, the depletion of either CD8+ or CD4+ T cells completely abolished the antitumor effect in *GPR171*LacZ/LacZ mice while the removal of NK cells greatly

**Fig. 3 T cells from GPR171 KO mice are hyperactive to antigen stimulation. a** GPR171 (β-gal) expression in thymus immune cells from naive WT or GPR171$^{+/LacZ}$ mice. **b** Flow cytometry analysis of GPR171 expression (β-gal) in activated CD4+ T cells from WT or GPR171$^{+/LacZ}$ mice. **c** Flow cytometry analysis of GPR171 expression (β-gal) in activated CD8+ T cells from WT or GPR171$^{+/LacZ}$ mice. **d** Flow cytometry analysis of GPR171 expression (β-gal) in splenic NK cells from WT or GPR171$^{+/LacZ}$ mice injected with poly I:C. **e** CFSE-labeled T cells from WT or GPR171$^{LacZ/LacZ}$ mice were stimulated with titrated mCD3 mAb for 3 days. Cell division of CD4+ and CD8+T cells were quantified by CFSE dilution. **f** CFSE-labeled WT or GPR171$^{LacZ/LacZ}$ T cells were stimulated with mCD3 mAb and BigLEN for 3 days. Cell division of CD4+ and CD8+ T cells were quantified by CFSE dilution. **g–i** Splenocytes from WT or GPR171$^{Lacz/LacZ}$ mice were transferred into B6D2F1/J to induce GVH response. Donor T cells in peripheral blood were determined by flow cytometry (**g**). On day 10 after transfer, the numbers of graft T cells in the spleen were enumerated (**h**). The alloreactivity of splenocytes was evaluated by their 4-h killing capacity against CFSE-labeled BALB/c splenocytes (**i**). **a** $n = 3$ biologically independent samples. **g**, **h** and **i** $n = 5$ biologically independent sample. Statistical significance was determined by one-way ANOVA for **a** and two-tailed Student's $t$-test for **a**, **g**, **h**, and **i**. Unless otherwise denoted, values are mean ± SEM. Source data was provided as a Source Data file. All data are representative of two independent experiments.

reduced the antitumor effect. Therefore, GPR171 ablation leads to enhanced antitumor immunity, which involves multiple immune cell types.

In a murine lung melanoma metastasis model, *GPR171*$^{lacZ/lacZ}$ and WT littermates were intravenously administered with $1 \times 10^5$ B16F10 melanoma cells. *GPR171*$^{lacZ/lacZ}$ mice displayed significantly longer overall survival than WT control mice (Fig. 4j). When we examined the mice for metastasis 4 weeks after tumor challenge, lung weight was significantly lower in *GPR171*$^{lacZ/lacZ}$ mice compared to littermate mice because of lower tumor burden (Fig. 4k, l); fewer tumor nodules were found in the lungs of *GPR171*$^{lacZ/lacZ}$ mice, compared to those from WT littermates (Fig. 4m). Consistently, 6 out of 8 WT mice exhibited tumor metastasis in other organs while we only detected metastasis other than lung in one *GPR171*$^{lacZ/lacZ}$ mouse ((Fig. 4n).

**Blockade of GPR171 signaling by an antagonist promotes antitumor immunity**. Next, we reasoned that GPR171 blockade by a GPR171 antagonist MS0021570, from a therapeutic perspective, would improve antitumor response. In MC38 colon cancer, the treatment of GPR171 antagonist right after MC38 inoculation significantly retarded tumor growth; as a result, 18 days after tumor inoculation, tumor weight in the GPR171 antagonist-treated group was only about 40% of the control group (Fig. 5a). Depletion of either CD4+ or CD8+ T cells markedly reduced the antitumor effect by GPR171 antagonist, supporting that GPR171 blockade-triggered antitumor immunity relies on both T-cell subsets (Fig. 5b). This is consistent with what we observed in *GPR171*$^{lacZ/lacZ}$ mice (Fig. 4i). When we delayed the treatment to 10 days post tumor inoculation, we were still able to see a significant delay of tumor growth (Supplementary Fig. 6a) by GPR171 antagonist, accompanied with extended mouse survival (Supplementary Fig. 6b).

We dissected tumor tissues to further determine the impact of GPR171 blockade on the TIME. We collected MC38 tumor tissues 17 days after tumor inoculation. GPR171 blockade led to a significant increase of CD45+ immune cells infiltrating the tumors; detailed analysis revealed that CD3+ T cells, mainly CD8+ T cells, were increased in tumors treated by GPR171 antagonist (Fig. 5c). As a result, the ratio of CD8+ to CD4+ T cells was significantly increased in tumors under the treatment of GPR171 antagonist (Fig. 5d). Though the density of intratumoral CD4+ Treg cells was not affected by GPR171 blockade, the percentages of CD4+ Treg cells in the GPR171 antagonist-treated group was greatly reduced (Fig. 5e). We further examined possible phenotypic changes in TILs under GPR171 blockade. Both CD4+ and CD8+ intratumoral T cells from mice with GPR171 blockade increased the expressions of PD-1 and TIGIT, two well-characterized immune checkpoints (Fig. 5f). Consistently, ex vivo stimulation and intracellular cytokine staining showed a significant increase in CD8+ T cells

expressing IFN-γ and TNF-α in the GPR171 antagonist-treated group (Fig. 5g).

To extend our observations to other mouse tumor models, we assessed the antitumor effect of GPR171 antagonist in mice subcutaneously implanted with B16-OVA or CT26 (colon cancer). In both tumor models, the treatment of GPR171 antagonist right after tumor cell inoculation was able to slow down tumor growth and thereby reduce tumor weight (Fig. 5h, i). In both tumor models, GPR171 blockade proportionally increased intratumoral CD3+ T cells and CD8+ T cells in CD45+ immune cells and increased the ratio of CD8+ to CD4+ T cells within tumors (Supplementary Fig. 6c, d). In CT26 tumors, GPR171 antagonist increased the percentages of IFN-γ and TNF-α-producing CD8+ T cells in response to the stimulation of AH1(Supplementary Fig. 6e), an H-2L$^d$-restricted peptide as the immunodominant antigen from CT26[36]. Although GPR171 signaling was reported to regulate feeding and anxiety behaviors[21,22], the weight and food intake in tumor-bearing mice under the treatment of GPR171 antagonist were not altered (Supplementary Fig. 6f). Taken together, our data suggest that GPR171 inhibition promotes a T-cell -mediated anti-tumor immunity.

**GPR171 blockade improves ICB therapy against cancer**. GPR171 blockade led to the upregulation of immune checkpoint receptors PD-1 and TIGIT on TILs (Fig. 5f), implying that these known immune checkpoints might limit the antitumor efficacy of the GPR171 antagonist. On the other hand, *GPR171* transcription in human melanoma was upregulated in response to immunotherapy[26,27]. In MC38 tumors, we confirmed that GPR171 expression in intratumoral T cells was significantly upregulated in tumors under the treatment of anti-PD-1 mAb (Fig. 6a). Thus, we hypothesized that GPR171 blockade would work synergistically with ICB to improve antitumor immunity. In a therapeutic mouse model of MC38, we started with the treatment of a PD-L1 mAb and GPR171 antagonist 10 days after tumor inoculation when all tumors had grown over 150 mm$^3$. The treatment of PD-L1 mAb or GPR171 antagonist alone modestly slowed down tumor growth while the combination of these two agents markedly inhibited tumor growth. As a result, half of the mice in the combination group survived over 45 days while all mice in the control group died within 20 days from the start of therapy (Fig. 6b). In mice with established CT26, we evaluated the combined antitumor effect of GPR171 antagonist with a TIGIT blocking mAb[37,38]. The delayed treatment of GPR171 antagonist or TIGIT mAb alone had minimal effects on slowing down tumor progression; however, GPR171 antagonist displayed a great synergy with anti-TIGIT to inhibit CT26 tumor growth, so as to extend mouse survival (Fig. 6c). Finally, we tested whether GPR171 blockade can sensitize immunotherapy in established B16F10 melanoma, a poorly immunogenic tumor

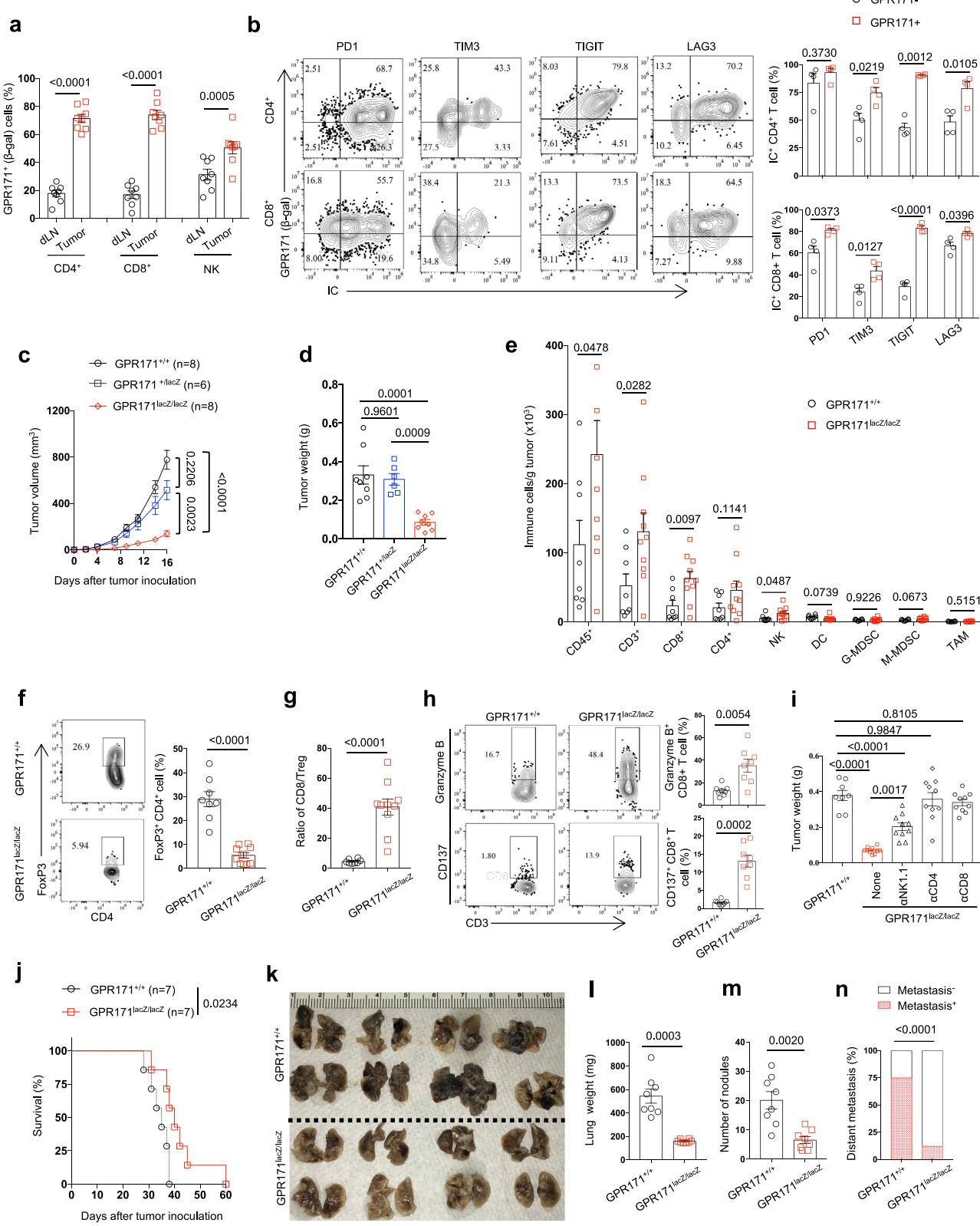

resistant to ICB therapy. GPR171 antagonist alone had no effect on tumor growth or mouse survival once B16F10 tumors were established. As expected, the treatment of ICB (PD-L1 and CTLA-4 mAbs) only displayed minor antitumor effects in established B16F10 tumors. However, the inclusion of GPR171 antagonist together with ICB significantly slowed down B16F1 tumor growth and extended mouse survival (Fig. 6d). When we dissected tumor tissues to examine TILs 14 days after therapy, intratumoral CD8+ T cells were proportionally increased only in the combination group (Fig. 6e); B16F10 tumors under the

**Fig. 4 Genomic knockout of GPR171 exhibits stronger antitumor immunity. a, b** GPR171^LacZ/lacZ mice were inoculated with MC38 tumors for 14 days. The expression of GPR171 (β-gal) on T cells and NK cells in dLN and tumor were quantified by flow cytometry (**a**). In GPR171^lacZ/lacZ tumors, TILs were examined for the expressions of GPR171 together with immune checkpoints (ICs), including PD-1, LAG3, TIGIT, and TIM3 (**b**). **c–h** GPR171^lacZ/lacZ GPR171^+/lacZ, and WT littermates were inoculated subcutaneously with MC38 tumor cells. Tumor growth curve (**c**) and tumor weight (**d**) at day 16 after tumor inoculation were determined. Single-cell suspensions were prepared from MC38 tumors to determine the densities of immune cells (**e**). The frequency of CD4+ Treg cells (**f**), the ratio of CD8/Treg (**g**), as well as the percentages of CD8+ T cells expressing Granzyme B or CD137 (**h**), within tumors were quantified. (**i**) MC38 tumor weights after 19 days of tumor inoculation were determined. In some mice, NK cells, CD4 + T cells, or CD8+ T cells were eliminated by corresponding depleting antibody. **j–n** GPR171^lacZ/lacZ and WT littermates were injected intravenously with B16F10 tumor cells. Overall survival of mice was followed till 60 days after tumor inoculation (**j**). 4 weeks later, gross images of lung metastasis were taken (**k**); lung weight (**l**) and tumor nodules in the lung (**m**) were quantified. The incidences of tumor metastasis in organs other than lung were recorded (**n**). **a** $n = 8$ biologically independent samples. **b** $n = 4$ biologically independent samples. **c, d** $n = 8$ (WT littermates and GPR171^lacZ/lacZ) and $n = 6$ (GPR171^+/lacZ) biologically independent samples. **e, i** $n = 8$ (WT littermates) and $n = 10$ (GPR171^lacZ/lacZ) biologically independent samples. **f–h** $n = 8$ (WT littermates) and $n = 8$ (GPR171^lacZ/lacZ) biologically independent samples. **j** $n = 7$ biologically independent samples. **k–n** $n = 7$ biologically independent samples. Statistical significance was determined by one-way ANOVA for **d** and **i**, two-tailed Student's $t$-test for **a, b, d, e, f, g, h, l** and **m**, two-way ANOVA for **c**, log-rank test for **j** or chi-square test for **n**. Unless otherwise denoted, values are mean ± SEM. Source data was provided as a Source Data file. Data are representative of two (**a, b, i, j, k, l, m,** and **n**) or three independent experiments (**c, d, e, f, g,** and **h**).

combinatory therapy contained the most CD8+ T cells producing effector cytokines IFN-γ and TNF-α (Fig. 6f). Taken together, our results support that blockade of GPR171 signaling improves ICB therapy against cancer in several mouse tumor models.

## Discussion

Our studies identify an unknown function for the GPCR receptor GPR171 in T-cell immunity. Disruption of GPR171 signaling promotes T-cell-mediated antitumor response and also improves ICB therapy. Therefore, our findings support that the BigLEN/GPR171 axis can be important in inhibiting tumor immunity.

GPR171 is a GPCR receptor closely related to the P2Y receptors, a group of GPCR receptors known to be important for immune response[39]. Early studies indicated that GPR171 is expressed on neuron cells to regulate food uptake[21]. The engagement of GPR171 by BigLEN triggers a G_{α15} or G_{i3} protein-mediated calcium flux[22]. Both GPR171 and its ligand BigLEN are highly conserved across human, mouse, and rat[21,40]. Our study here demonstrated that GPR171 is inducible in T cells of both human and mouse, and GPR171 engagement by BigLEN inhibits TCR-mediated signaling cascades, including calcium flux, NFAT activation, and PLC-γ1/ERK phosphorylation. Knockout of GPR171 in Jurkat cells completely abolished the suppressive effect on TCR signaling by BigLEN, suggesting that BigLEN- triggered inhibitory effect is mediated through GPR171. The GPR171-mediated T-cell suppression was further confirmed in studies of GPR171^LacZ/LacZ mice. To our knowledge, this is the first study that has demonstrated a crucial role of GRP171 in T-cell immunity. It is still unclear how GPR171 signaling has a completely opposite effect on calcium flux between T cells and neuron cells. In neurons, GPR83 interacts with GPR171 to regulate BigLEN- mediated signaling[41], and the lack of GPR83 in conventional T cells could possibly lead to a different outcome of GPR171 signaling[42]. In addition, the possibility that GPR171 operates through different G proteins in different cell types is currently under investigation.

Our study supports GPR171 as a T-cell checkpoint important for tumor immunity. The kinetics of GPR171 expression in T cells are very similar to other immune checkpoints, such as PD1, TIGIT, CD112R, and TIM3[6]. Though GPR171 is transcribed in naive T cells, our X-gal or β-gal staining was unable to detect GPR171 in naive mouse splenic T cells, suggesting a tightly post-transcriptional regulation of GPR171. Upon stimulation, activated T cells upregulate the expression of GPR171. In both human and mouse, GPR171 is enriched in TILs. Blockade of GPR171 prevents peptide-induced T-cell anergy and improves antitumor immunity. Consistently, T cells from GPR171^lacZ/lacZ mice are hyperreactive to TCR stimulation and GPR171^lacZ/lacZ mice are resistant to tumor challenge. Though the role of GPR171 in human cancer has not been particularly investigated, we are able to reveal from early clinical trials that GPR171 expression in cancer is upregulated in response to immunotherapy. In patients with metastatic melanoma under ipilimumab (anti-CTLA4) therapy, GPR171 expression in melanoma was considerably increased upon ipilimumab and its expression is significantly higher in responders than non-responders[27]. In a cancer vaccine trial, the expression of GPR171 in tumors was upregulated upon MAGE-A3 vaccine and its expression is positively associated with clinical response outcome in patients with metastatic melanoma[26]. All these warrant a further comprehensive investigation into the role of GPR171 in ICB efficacy of cancers in the clinic.

BigLEN is a neuropeptide derived from proSAAS (encoded by PCSK1N), a neuroendocrine peptide precursor predominantly expressed in the central nervous system (CNS)[40]. Besides neuron cells in the CNS, proSAAS is also produced by endocrine cells within peripheral tissues[43]. The regulation of BigLEN expression or secretion outside of the CNS, particularly in the TIME, is unclear yet. One research group has been able to detect glycosylated BigLEN from both mouse and human islets by targeted mass spectrometry[44]. Given that overactive T-cell immunity is seen in our GPR171^LacZ/LacZ mice, it is tempting to speculate whether altered immune functions are also present in proSAAS^−/− mice, presumably due to the lack of the ligand BigLEN[45]. Our identified role for the BigLEN/GPR171 axis in T-cell immunity supports that this pathway could contribute to the immune privilege in the CNS. Besides the physical blood-brain barrier (BBB), the CNS executes several immunosuppressive mechanisms to actively contain excessive neuroinflammation[46]. During neuroinflammation, pathological T cells travel into the CNS to execute their functions, and the presence of BigLEN would presumably engage GRP171 to prevent the overreaction of intracerebral T cells. Therefore, quantification of BigLEN in these inflamed tissues should better reveal the role of BigLEN in immune response in vivo. Finally, it is unknown whether there are additional ligands to interact with GPR171 other than BigLEN. A recent publication indicated that GPR171 can respond to lipid fractions of bacterial extracts[47]. In this scenario, our observations of enhanced T-cell immunity in GPR171^LacZ/LacZ mice, as well as the improved antitumor effect of GPR171 antagonist might not be solely through BigLEN blockade. When we treated mice with a broad antibiotic to eliminate gut microbiota, the antitumor effect of GPR171 antagonist seemed not to be disrupted (Supplementary Fig. 7). Together, it remains to be determined the source of GPR171 ligands that contributes the suppression of antitumor immunity.

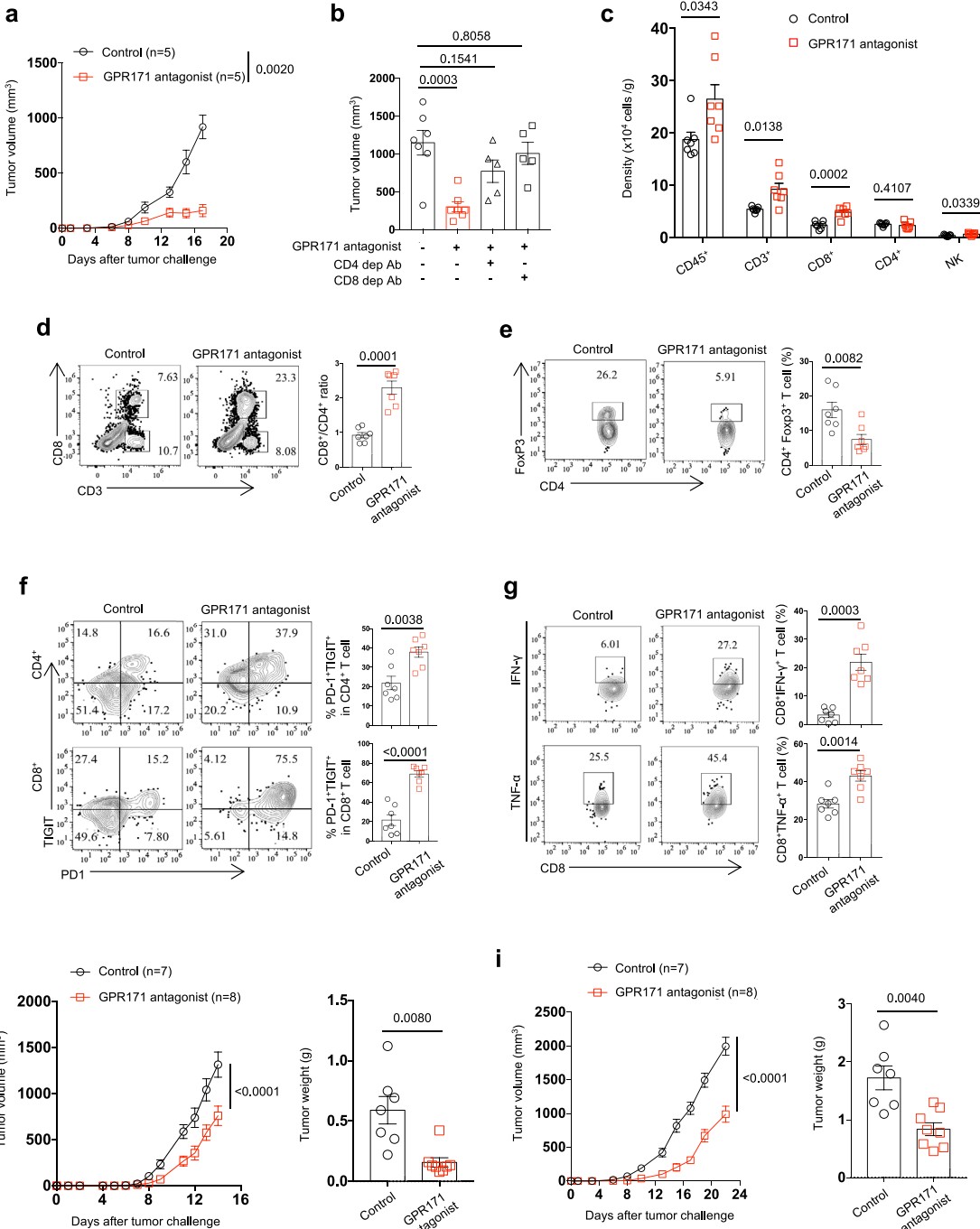

**Fig. 5 GPR171 antagonist promotes T-cell-mediated antitumor immunity. a**, **b** Black B6 mice were inoculated with MC38 tumor cells and followed with the treatment of GPR171 antagonist or control saline three times per week. Tumor volume was recorded (**a**). In some mice, CD4+ or CD8+ T cells were depleted by specific antibodies right before tumor challenge; tumor volume at 18 days after tumor inoculation was determined (**b**). **c**–**g** Single-cell suspensions were prepared from MC38 tumors harvested on day 17 after tumor inoculation. Flow cytometry analysis was performed to determine the densities of infiltrating immune cells (**c**). The ratio of CD8+ to CD4+ T cells (**d**), the percentages of Foxp3+ Treg in CD4+ T cells (**e**), as well as PD-1 and TIGIT-double-positive CD8+ T cells (**f**) were determined by flow cytometry analysis. The percentages of IFN-γ and TNF-α -producing cells in CD8+ TILs were determined by intracellular staining (**g**). **h** Black B6 mice inoculated with B16-OVA tumor cells were treated with GPR171 antagonist or control saline daily right after tumor inoculation. Tumor volume and weight at 14 days after tumor inoculation were recorded. **i** BALB/c mice inoculated with CT26 tumor cells were treated with GPR171 antagonist or control saline daily right after tumor inoculation. Tumor volume and weight at 21 days after tumor inoculation were recorded. **a** $n = 5$ biologically independent samples. **b** $n = 7$ (control and GPR171 antagonist group) and $n = 5$ (GPR171 antagonist + CD4 mAb and GPR171 antagonist + CD8 mAb group). **c**–**g** $n = 7$ biologically independent samples. **h**, **i** $n = 7$ (Control) and $n = 8$ (GPR171 antagonist). Statistical significance was determined by One-way ANOVA for **b**, two-tailed Student's t-test for **c**, **d**, **e**, **f**, **g**, **h** and **i**, Two-way ANOVA for **a**, **h**, and **i**. Unless otherwise denoted, values are mean ± SEM. Source data was provided as a Source Data file. Data are representative of three (**a**, **b**, **c**, **d**, **e**, **f**, and **g**) or two independent experiments (**h** and **i**).

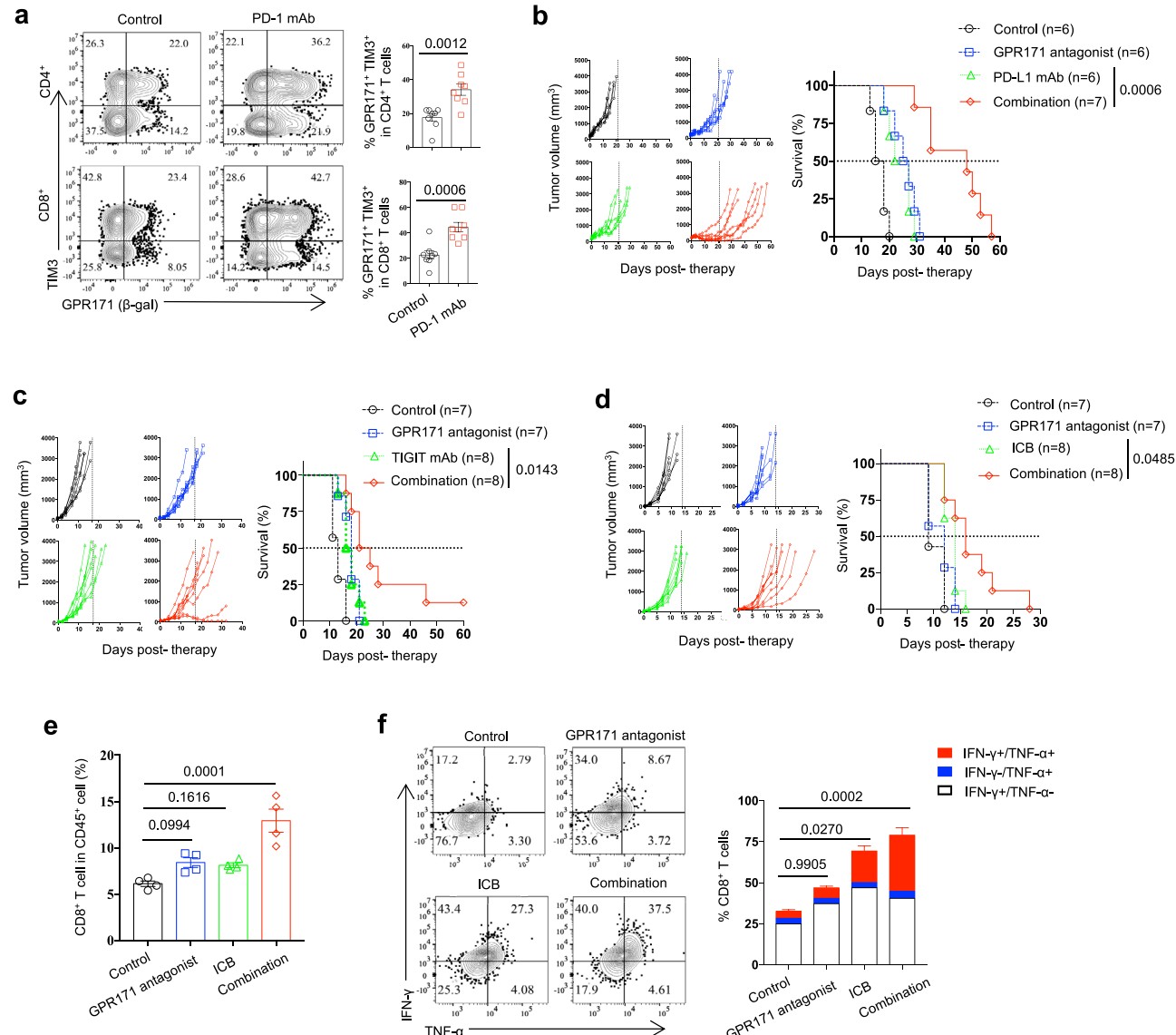

**Fig. 6 GPR171 blockade improves therapeutic antitumor effect of ICB. a** TILs from MC38 tumors in GPR171+/LacZ mice under 10 days control or PD-1 mAb treatment were analyzed for TIM3 and GPR171 expression. **b** Tumor growth and tumor-related survival of MC38-bearing mice under the treatment of PD-1 mAb and GPR171 antagonist were shown. Therapy was started 10 days after tumor inoculation. **c** Tumor growth and tumor-related survival of CT26-bearing mice under the treatment of TIGIT mAb and GPR171 antagonist were shown. Therapy was started 7 days after tumor inoculation. **d–f** Tumor growth and survival of B16F10-bearing mice treated with ICB (PD-L1 + CTLA-4 mAbs) together with GPR171 antagonist were shown (**d**). Therapy was started 7 days after tumor inoculation. 14 days after therapy, single-cell suspensions were prepared to determine the percentages of intratumoral CD8+ T cells (**e**) and cytokine- producing CD8+ T cells (**f**). In Figure **b**, **c**, and **d**, the dash line in each left panel indicates the date when all control mice were dead due to tumor burden. The dash line in each right panel shows median survival. GPR171 antagonist was administrated three times a week for up to total nine times while antibody was injected twice per week for up to total six times. **a** $n = 8$ biologically independent samples. **b** $n = 6$ (control, GPR171 antagonist and PD-L1 group) and $n = 7$ (combination group). **c** $n = 7$ (control, GPR171 antagonist and TIGIT group) and $n = 8$ (combination group). **d** $n = 7$ (control and GPR171 antagonist group) and $n = 8$ (ICB and combination group). **e** and **f** $n = 4$ biologically independent samples. Statistical significance was determined by One-way ANOVA for **e** and **f**, two-tailed Student's t-test for **a**, or Log-rank test for **b**, **c** and **d**. Unless otherwise denoted, values are mean ± SEM. Source data was provided as a Source Data file. All data are representative of two independent experiments.

In conclusion, our study identified an immunomodulatory role of the BigLEN/GPR171 pathway in T cells. GPR171 blockade promotes antitumor immunity to inhibit tumor progression and offers a target for cancer immunotherapy.

## Methods

**Chemicals and antibodies.** Human BigLEN peptide (Catalog #004-53) was purchased from Phoenix Pharmaceuticals Inc (Burlingame, CA). Mouse BigLEN peptide (Catalog #6304) and a mouse GPR171 antagonist (MS0021570, Catalog #6298) were purchased from Tocris Bioscience (Minneapolis, MN). Jurkat cells stably expressing a luciferase reporter under the control of NFAT response element (Jurkat-NFAT-Luc) have been established[32]. OVA257-264 peptide (SIINFEKL) was synthesized by GenScript (Piscataway, NJ). Antibodies for in vivo usage, including anti-mouse CD8β mAb, anti-mouse CD4 mAb, anti-mouse CTLA-4 mAb and anti-mouse TIGIT mAb were purchased from BioXcell (West Lebanon, NH). The detailed information of these antibodies was included in supplementary table 1. Hybridoma for mouse PD-L1 (B7-H1) neutralizing mAb (clone 10B5) was obtained from Dr. Lieping Chen's laboratory at Yale University[48]. GP70 (423–431, AH1) peptide was kindly provided by Dr. Jill Slansky (Department of Immunology and Microbiology, University of Colorado Anschutz Medical Campus (CU AMC)). Phorbol 12-myristate13-acetate (PMA) (Catalog #79346) and Ionomycin (Catalog #I9657) were purchased from Sigma-Aldrich (St. Louis, MO).

**Cell lines**. Mouse tumor cell lines MC38 (colon adenocarcinoma), B16F10 and B16-OVA (melanoma), and Jurkat cells (human T-cell lymphoblast) were cultured and maintained in the laboratory. CT26 (colon adenocarcinoma) was from Dr. Jill Slansky's laboratory. All these cell lines were tested for Mycoplasma infection by DAPI staining upon purchase or obtaining. Tumor cells were cultured in DMEM or RPMI1640 containing 10% FBS and 1% penicillin/streptomycin. *GPR171* gene in Jurkat cells was knocked out by the CRISPR/Cas9 technology. Briefly, Jurkat cells were infected with lentivirus with Cas9-single guide RNAs (sgRNA) targeting *GPR171*. Cells were incubated with 1 μg/ml puromycin for 10 days to enrich puromycin-resistant cells. After limiting dilution, *GPR171*KO Jurkat cells were selected by western blot analysis.

**GPR171lacZ/lacZ mice generation and GPR171 (β-galactosidase) detection**. The *GPR171*lacZ/lacZ mice used in this study were generated from an embryo stem cell clone (Deletion, clone AE8) from the Knockout Mouse Project (KOMP) Repository[49]. Blastocyst injections were performed, and germline transmission was established at National Jewish Health Animal Core Facility. Littermates were used for phenotypic studies of *GPR171*lacZ/lacZ mice. For X-Gal staining of isolated immune cell suspensions and bulk tissues from *GPR171*+/LacZ mice, β-galactosidase reporter gene staining kit (Catalog #GALS, SIGMA) was used under recommended protocol. After X-Gal staining, bulk tissues were stained with eosin to evaluate microscopic images of X-gal signal. For flow cytometry staining of β-galactosidase, single-cell suspensions were stained for surface molecules before fixed by methanol and stained with β-galactosidase antibody.

**Reverse transcription PCR (RT-PCR) and quantitative PCR**. RNA was extracted using RNAqueous® Micro Kit (Catalog #AM1931, Invitrogen), and complementary DNA (cDNA) was synthesized using PrimeScript™ RT-PCR Kit (Catalog #PR014, Takara Bio). Dream Taq™ Hot Start PCR Master Mix (Catalog #K9011, Thermo Fisher Scientific) was used to detect GPR171 gene. Annealing temperature in each primer set was determined by Tm calculator (Thermo Fisher Scientific). The information of primer sequences was provided in Supplementary Table 2. Images were obtained and quantified via Chemi Doc XRS+ and Image Lab Software (Bio-Rad). qPCR was carried out by the QuantStudio™ 5 Real-Time PCR System in a 96-well plate (Applied Biosystems). Relative quantification of mRNA expression of each gene was performed by gene-specific primers together with the PowerUp™ SYBR™ Green master mix (Catalog #A25741, Applied Biosystems). Expression levels of GAPDH as a housekeeping gene ($CT_{control}$) and target genes ($CT_{target\ gene}$) were determined. Normalizing expression level of target genes ($\Delta CT_{target\ gene}$) was calculated ($CT_{target\ gene} - CT_{control}$). The expression fold change of target genes was calculated ($2^{-\Delta CT_{target\ gene}}$) in each triplicate sample.

**In vitro T-cell stimulation**. CD4+ T cells were isolated from WT B6 mice by mouse CD4+ T-cell isolation kit (Catalog #130-104-454, Miltenyi Biotec) and were stimulated by 96-well plate-coated mouse CD3 mAb (Clone 145-2C11, Catalog #100340, Biolegend). OT-1 splenocytes were stimulated by OVA257-264 peptide. We isolated either CD4+ or CD8+ T cells from human peripheral blood mono-nuclear cells (PBMC) by human CD4+ or CD8+ T-cell isolation kit (Catalog #130-096-533 and #130-096-495, Miltenyi Biotec) and stimulated them with plate-coated human CD3 mAb (Clone OKT3, Catalog #317326, Biolegend). To track T-cell division, T cells were labelled with CFSE (Biolegend, 2 μM) before stimulated with CD3 mAb with or without different dosages of mouse or human BigLEN for 6 days. Supernatants from CD4+ T-cell culture were collected on day 1 and 6 to detect cytokines. The concentration of mouse IL-2 in one-day cultured super-natants was measured by ELISA (Catalog #555148, BD Biosciences). Other cyto-kines in 6-day cultured supernatants were determined by LEGEND plex™ mouse Th17 panel (Catalog #741047) or human Th17 panel (Catalog #740728, Biolegend), according to the Manufacturer's instructions.

**Western blot**. For western blot, proteins were extracted from PBMC, WT or GPR171KO Jurkat cells by M-PER™ Mammalian Protein Extraction Reagent (Catalog #78501, Thermo Fisher Scientific) with protease and phosphatase inhibitors (Catalog #4906845001, Roche). Polyclonal human GPR171 antibody was generated by ABGENT (SanDiego, CA) by immunizing New Zealand Rabbits with a GPR171 peptide (TETFASPKETKAQK, 297-310aa) conjugated to KLH. Phosphorylated PLCγ-1, PLCγ-1, phosphorylated p44/42 (ERK1/2), p44/42, phosphorylated AKT, AKT, phosphorylated ZAP70, ZAP70, phosphorylated CD3ζ and β-actin antibodies were purchased from Cell Signaling Technology. CD3ζ antibody was purchased from Santa Cruz Biotechnology. Detailed information of these antibodies for western blot were included in supplementary table 3. We used TGX FastCast Acrylamide Kit, 12% for electrophoresis and Trans-Blot Turbo RTA Midi 0.2 μm Nitrocellulose Transfer Kit (Bio-Rad) for transferring proteins to nitro-cellulose membranes. After protein transfer, the membrane was incubated with 5% milk for 30 min at room temperature and then incubated with primary antibody overnight. Next day, the membrane was incubated with the peroxidase-labeled secondary antibody (Catalog #7074 S, Cell Signaling Technology) for 2 h after washed with TBS-T washing reagent. Finally, the protein bands were detected by Pierce® ECL Western Blotting Substrate (Catalog #32106, Thermo Fisher

Scientific). Images were obtained and quantified via Chemi Doc XRS + and Image Lab Software (Bio-Rad).

**Intracellular Ca2+ flux and luciferase assay**. Intracellular $[Ca^{2+}]$ in Jurkat cells and human PBMC was measured by Fluo-4 Calcium Imaging Kit (Catalog #F10489, Thermo Fisher Scientific). Briefly, cells were incubated with Fluo-4 AM loading solution at 37 °C for 15 min followed by 15 min at room temperature. After incubation, cells were washed and prepared by HBSS with calcium, magnesium and 2% FBS. OKT3 (anti-human CD3 mAb, 0.5 μg/ml) was added to stimulate Jurkat-NFAT-Luc cells at 15 s after the starting of intracellular $[Ca^{2+}]$ analysis by FACS Calibur (BD Bioscience, Franklin Lakes, NJ, USA). Human PBMC was stimulated by biotin- labeled OKT3 (5 μg/ml) together with streptavidin (20 μg/ml) for crosslinking. BigLEN was added 10 min before the analysis. The MFI (Fluo-4) at each 5 s was determined and then the area under curve (AUC) was calculated to evaluate the effects of BigLEN to CD3 stimulation[50]. For luciferase Assay, WT or GPR171KO Jurkat-NFAT-Luc cells were co-cultured with CHO stimulator cells stably expressing membrane-bound OKT3 and PD-L1 (PD-L1 + CHO stimulator)[32] at the ratio of 10 to 1 for 4 h with or without human BigLEN. Cells were lysed with the ONE-Glo Luciferase Assay System (Catalog #E6120, Promega) and measured for luminescent signal instantly.

**OT-1 transgenic T-cell response vivo**. Naïve CFSE-labelled (5 μM) CD45.1+ OT-1 T cells at $5 \times 10^6$/mouse were intravenously transferred into C57BL/6 mice, which were inoculated with $2 \times 10^5$ B16-OVA tumor cells 7 days before. GPR171 antagonist (3 mg/kg) or vehicle was intraperitoneally injected on day 0, 1, 3, and 6 after OT-1 T-cell transfer. On day 5 and 7, the divisions and percentage of transferred OT-1 CD8 T-cell (CD45.1+) in both drainage lymph node and spleen were analyzed by flow cytometry after single-cell isolation.

In a peptide-induced anergy model, C57BL/6 mice intravenously transferred with $3 \times 10^6$ naïve OT-1 T cells (CD45.1+) were challenged with OVA (257–264) peptide (0.1 mg per mouse) intravenously. GPR171 antagonist (3 mg/kg) or vehicle was intraperitoneally injected both 1 and 3 days after peptide injection. On day 4 and 7, blood samples were collected from a tail vein to determine the percentages of transferred OT-1 CD8+ T cells. We also evaluated the total number of OT-1 CD8+ T cells in the spleen on day 7.

**Mouse tumor models**. Wild type C57BL/6 (Stock #000664) and BALB/c (Stock #000651) mice were purchased from the Jackson Laboratory (Bar Harbor, ME). All mice including OT-1 TCR transgenic mice were maintained in a specific pathogen-free environment at the Office of Laboratory Animal Research at the University of Colorado Anschutz Medical Campus, where they were maintained 21 °C on a light-dark cycle (6 a.m. to 8 p.m.) and given free access to food and water. Six- to 10-week-old mice of both sexes (same sex per experiment) were used in this study.

MC38 ($5 \times 10^5$), CT26 ($5 \times 10^5$), B16 ($2 \times 10^5$), or B16-OVA ($2 \times 10^5$) cells were subcutaneously injected into the right flank of syngeneic C57BL/6 or BALB/c mice. Mice were randomized into different treatment groups. The GPR171 antagonist MS0021570 at 3 mg/kg was injected intraperitoneally three times a week. We used two different tumor models, prevention and therapeutic, to assess the antitumor effect of GPR171 blockade. In the prevention model, mice were started with the treatment the day after tumor inoculation. For the therapeutic model, we started the treatment between 7 to 10 days after tumor inoculation, when implanted tumors became measurable in all mice. Monoclonal antibodies (mAbs) against known immune checkpoints, including PD-L1 (200 μg/mouse), TIGIT (200 μg/mouse) and CTLA-4 (200 μg/mouse) were administered twice per week over 2 weeks. These antibodies were diluted in 200 μl of PBS. To evaluate the effects of GPR171 ligands derived from microbiome in the gut, B6 mice were feed with antibiotic drinking water, which includes amoxicillin (1 mg/ml), streptomycin (5 mg/ml) and colistin (1 mg/ml) for 2 weeks before inoculated with MC38 subcutaneously. Tumors (length × width) and body weight were measured every 2 or 3 days with a caliper and tumor volume was determined as 1/2 × (length × width²). Mice were euthanized if the tumor length reached 20 mm or the tumor became ulcerated. We established a melanoma lung metastasis model in *GPR171*lacZ/lacZ and littermates by injecting $1.0 \times 10^5$ B16F10 cells through the tail vein. Mice were euthanized 28 days after B16 tumor challenge to evaluate tumor burden in the lung. The method of euthanasia was performed in accordance with the US Department of Health and Human Services Guide for the Care and Use of Laboratory Animal. Carbon dioxide was administered from a compressed gas tank for 2 min. A secondary physical euthanasia was added by cervical dislocation. Our animal protocols (00132 and 00461) were approved by the Institutional Animal Care and Use Committee of the University of Colorado Anschutz Medical Campus.

**Mouse GVHD model and allogenic CTL assay**. To induce acute GVHD, $5 \times 10^7$ splenocytes from C57BL/6 (H-2b) mice (WT or GPR171lacZ/lacZ mice) were injected intravenously into B6D2F1/J (H-2b+d) mice (Stock #100006), which were purchased from the Jackson Laboratory on Day 0. The percentages of donor CD4+ and CD8+ T cells were determined in the peripheral blood on day 5, 8, and 10. On day 10, splenocytes were isolated to quantify donor CD4+ and CD8 T+ cells in the spleen. Their allogeneic cytotoxic activities were determined by co-culturing them

with BALB/c ($H-2^d$) or C57BL/6 splenocytes ($H-2^b$) for 4 h to measure the death of target cells by propidium iodide (PI) staining.

**Single-cell isolation and flow cytometry**. Mouse tumor tissues were collected, cut into small pieces and re-suspended in RPMI-1640 medium with Liberase[TM] (Catalog # 5401117001, Roche Diagnostics Corporation, Indianapolis, IN). Tumors were digested for 50 min in a shaking incubator at 37 °C. After centrifugation, digested tissues were passed through a 100-μm cell strainer to make single-cell suspensions. For isolating intraepithelial lymphocytes from mouse small intestine, the tissues were incubated with dithioerythritol (DTE) solution (1 mM) on 37 °C and a magnetic stirrer for 30 min. Digested tissues with the supernatant were passed through a 100-μm cell strainer and then centrifuging at $500 \times g$ for 5 min. Lamina propria was removed from the wall of small intestine before tissue digestion. Single-cell suspensions were blocked with LEAF[TM] anti-mouse CD16/32 for 20 min on ice before staining with primary conjugated antibodies. For intracellular staining of cytokines, Monensin solution (Catalog #420701, Biolegend) was added during ex vivo T-cell stimulation. For staining FoxP3, the samples were prepared by eBioscience[TM] FoxP3/Transcription Factor Staining Buffer Set (Catalog #00-5523-00) after surfaced markers staining. Detail information of the antibodies for flow cytometry was updated in supplementary table 4. Samples were analyzed by a CytoFLEX (Beckman Coulter, Indianapolis, IN, USA). Data was analyzed using FlowJo software (Tree Star).

**Immune cells sorting and total RNA isolation**. Immune cell subsets were sorted from human PBMC (CD3+, CD4+, CD8+, CD19+, CD14+, CD56+), mouse splenocytes (CD3+, CD4+, CD8+, CD19+, NK1.1+) and thymocytes (CD4+ and CD8+, CD4+ and CD8−, CD4−, and CD8+, CD4−, and CD8−) by MoFlo XDP70 (Beckman Coulter) at flow cytometry core of the University of Colorado Cancer Center. After sorting, Total RNA of each cell type was isolated and then cDNA was synthesized in the same day. These cDNA samples were used for RT-PCR and qPCR to determine the expression of GPR171.

**Data and statistical analysis**. All statistical analyses were performed using IBM SPSS statistics 24 version software for Mac and GraphPad Prism 7.0 software (GraphPad Software). The two-tailed Student's $t$-test or one-way analysis of variance (ANOVA) was performed to compare differences in the non-parametric variables between the groups. The log-rank test was performed to compare the survival and the two-way ANOVA was performed to compare time-dependent tumor growth. Each sample in vitro was analyzed in more than triplicate samples. All $P$-values < 0.05 were considered to be significant.

**Reporting summary**. Further information on research design is available in the Nature Research Reporting Summary linked to this article.

## Data availability

All numerical source data underlying Figs. 1–6 and Supplementary Figs. 1–7 are provided as a Source Data file. We used public databases to evaluate GPR171 transcript. DOI or URL in each database was shown. Data presented in Supplementary Figs. 1a and 3a were derived from the BioGPS database (http://biogps.org/#goto=welcome). GPR171 expression data in human cancers (Supplementary Fig. 1b) were extracted from the TCGA database (https://portal.gdc.cancer.gov). The scRNAseq dataset for human melanoma (Supplementary Fig. 1c) is at https://doi.org/10.1126/science.aad0501 and the dataset for liver cancer (Supplementary Fig. 1d, e) is at https://doi.org/10.1016/j.cell.2017.05.035. Source data are provided with this paper.

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

## Acknowledgements
We thank the Mouse Genetics Core Facility at National Jewish Health for generating GPR171$^{lacZ/lacZ}$ mice, the flow cytometry core at the University of Colorado Cancer Center and the Support Grant (P30CA046934) for cell sorting, and the ImmunoMicro Flow Cytometry Shared Resource Laboratory at CU AMC (RRID: SCR_021321) for flow cytometry analyses. We also thank Dr. Haruhiko Furusawa at Department of Medicine at CU AMC for microscopic X-gal staining, Dr. Yi Zhang at Temple University for comments. This study is partially supported by the Wings of Hope for Pancreatic Research and the Research Scholar Grant, RSG-17-106-01 LIB, from the American Cancer Society.

## Author contributions
Y.F., R.D.S., and Y.Z. designed the study; Y.F. and Y.Z. developed methodology of the study. Y.F., R.J.T., Y.S., E.N.M., N.B., and F.H. acquired the data. Y.F., T.W., R.M.T., W.Z. and Y.Z. analyzed and interpreted the data. Y.F. and Y.Z. prepared the manuscript for submission and revision. R.D.S. and Y.Z. supervised the study; all authors approved the final manuscript.

## Competing interests
The authors declare no competing interests.
