## [Peer Review File · Nature Communications]

REVIEWER COMMENTS

Reviewer #1 (Remarks to the Author):

In the article entitled "The BigLEN/GPR171 pathway suppresses T cell activation and limits antitumor immunity" by Fujiwara and colleagues, the authors describe a novel role for the G protein-coupled receptor GPR171 in regulating T cell-mediated immunity. While a role for BigLEN/GPR171 has been described to regulate food uptake and anxiety, the role for GPR171 in T cells has not been characterized. Here, the authors show that GPR171 acts as an inhibitory receptor on T cells, suppressing T cell receptor signaling, proliferation, and cytokine production. Blocking GPR171 promotes productive anti-tumor T cell responses as a single agent, but has minimal impact on tumor growth, likely because of compensation from other inhibitory receptors. Co-blockade of GPR171 and either PD-1 or TIGIT resulted in enhanced tumor control compared to either single agent treatment, highlighting the potential for GPR171 blockade to augment pre-existing checkpoint blockade modalities. Overall, the authors provide convincing evidence for the role of GPR171 as a negative regulator T cell functions in a variety of different models. The experiments seem well executed and the conclusions are supported by the data provided. There are some points outlined below which need to be addressed prior to publication, which are intended to strengthen an already well executed study.

Major Points:

- 1.) The most important point that needs to be addressed prior to publication is reporting on the number of independent experiments for each of the results included in the paper. The n within each figure is provided in the Figure Legends or obvious from the way the data is displayed, but the number of independent experiments performed seems to be missing. In order to evaluate the reproducibility of the work, this information needs to be included, preferably in the Figure Legends.
- 2.) Considering the authors report on a new inhibitory receptor pathway, it would be useful to place this new receptor in context with the other inhibitory receptors. Work from John Wherry's group (Blackburn 2009 PMID 19043418) has shown that in chronic LCMV, many of the inhibitory receptors tend to be co-expressed on the same cells. Can the authors provide information on whether GPR171 also tends to be co-expressed with the other inhibitory receptors, particularly in the tumor setting? There are some data provided on co-expression with Tim-3 (Figure 6A), but inclusion of co-expression with PD-1 and/or TIGIT would also be useful since these are the two inhibitory receptors that the authors follow up on with in vivo blockade.
- 3.) Can the authors provide a more in depth discussion on the ligand BigLEN, including which cell types produce it, and how this may impact the ability of the BigLEN/GPR171 pathway to regulate T cell functions? Is there any information in the literature/publically available data sets on BigLEN expression outside of the CNS?
- 4.) The authors mention that the BigLEN/GPR171 has mostly been described in the context of regulating feeding and anxiety behaviors. Did the authors observe any impact on these aspects in their studies blocking the GPR171 pathway? A discussion of this aspect of the pathway would be useful to add, especially when considering adding GPR171 blockade to pre-existing immunotherapy regimens. Would issues with feeding/anxiety be a type of "immune-related adverse event" to consider when thinking about immunotherapy in patients.
- 5.) The authors provide data on GPR171 in both mice and humans, as well as some comparison of expression patterns. Can the authors comment on the amount of homolog between the mouse and human receptors? Is this pathway highly conserved between mice and humans?
- 6.) The authors provide frequencies of immune cell subsets relative of total in MC38 and B16F10, and there seems to be increased CD8 T cells and decreased Treg cells. Can the authors provide the CD8/Treg ratios following GPR171 modulation in these models? This is a useful benchmark for thinking about the protective vs. immunosuppressive microenvironments.

Minor Points:

1.) In Figure 4G, there should be a label on the data point colored in red, or an explanation of what it is in the Figure Legend. Presumably it is GPR171 lacZ/lacZ without any treatment (or vehicle treated), but this should be indicated in the Figure.

2.) The resolution of Figure S1A is very poor, it should be increased to make the labels more legible.

Reviewer #2 (Remarks to the Author):

This article describes a role for GPR171 in T cell activation, with evidence that signaling through GPR171 (e.g. by the neuropeptide BigLEN) has a suppressive influence on signaling through the TCR. These findings are taken further to show that GPR171 inactivation leads to stronger T cell activation, and that this can be beneficial in an anti-tumor response.

Overall, this is an exciting and novel finding, of potential translational importance. My major reservation is that there is no evidence that any of these experiments has been performed more than once. For example, for Fig 1, the implication is that they were really only done once: "...PBMCs of a healthy donor...". There are similar issues with all of the data. Indeed, most of the legends don't mention the n for the (technical or biologic) replicates. This is a major issue. Another issue is the statistics, which are mostly not described. For Fig. 4A (left panel) as a – not unique – example: the SEM error bars overlap, yet the p value is given as <0.01. The test is not defined, and frankly, a "significant" p value is a little difficult to believe.

Fig 1 shows data on human T cell expression, where the most expression is found in immune cells with CD8>CD4>everything else. Thymic expression is low. These data are from BIOGPS. Stimulation with aCD3 plus BigLEN reduces signaling by various methods, and the GPR171 gene is expressed in T cells and NK cells in TILs. The signaling reduces Ca²⁺ flux and various cytokine's expression. As noted, it looks like each experiment was only done once. There should also be a control for BigLEN treatment without TCR stimulation. The level of the GPR171 effect on TCR stimulation should be defined more. Is it at the level of CD3, ZAP70 or LAT or later?

The BIOGPS data for GPR171 expression of in mouse T cells are not shown. Strangely, they are very different to the human data (<http://biogps.org/#goto=genereport&id=229323>), with mast cells>DC>NK>B>T, and rather low expression in thymus. This discrepancy is not mentioned. The PCR data (suppl 3A) are not quantitative and therefore not very useful. This said, the data on inhibition of OT-I CD8+ cell and CD4+ cell stimulation look good (how many times were they done?).

Using the antagonist of GPR171, MS0021570, they find that the T cells respond better to TCR stimulation, and there is more expansion in tumor-bearing mice.

The authors made a knockin mouse for the GPR171 locus that replaced the coding part of GPR171 with LacZ. GPR171+/LacZ mice show lots of expression of LacZ in the thymus, more in the early DN thymocytes than in the DP or SP subsets. BIOGPS did not show this (DN not shown, DP lower than SP), so I think there should be more dissection of the expression in normal mice.

The "knockout" mice (LacZ/LacZ) responded better than LacZ/+ mice to TCR stimuli, and LacZ/+ responded worse than WT when BigLEN was added to stimulate the GPR171. So, GPR171 allows suppression of TCR signals by BigLEN. In antitumor responses, the LacZ/LacZ "knockout" worked better, and depletion of CD4, CD8 or NK cells all reduced the effect of higher activity in the LacZ/LacZ mice.

The antagonist MS0021570 allowed more TCR signaling, resulting in higher expression of checkpoint inhibitory molecules PD-1 and TIGIT. aPD-1 caused upregulation of GPR171 on TILs. There are intriguing data showing that MS0021570 works synergistically with different checkpoint blockade antibodies (best was a combination of aPD-L1 + aCDTA4).

The data are good and interesting, but they do not appear to have been replicated.

Minor points:

OT-I, not OT1.

BALB/c, not Blab/c or Balb/c.

Repetition in Discussion

1st para:

"GPR171 expression is inducible in both human and mouse T cells, and amplification of GPR171 signaling via BigLEN delivers a suppressive signal to T cells."

2nd para:

"GPR171 is inducible in T cells of both human and mouse, and GPR171 engagement by BigLEN inhibits TCR-mediated signaling cascades."

Suppl Fig 3D scale needs to be "broken" or changed to log, since most of the cytokines are expressed too weakly to see the differences (apparently very statistically sig) on the graph.

Reviewer #3 (Remarks to the Author):

Fujiwara et al studied how GPR171 regulates T cell function in relation to T-cell immune checkpoint activity. The authors used their in vitro results to inform specific in vivo experiments in order to study the role of GPR171 in the tumor microenvironment and how it may act as a novel immune checkpoint. The GPR171 is a GPCR which, until recently was classified as an orphan receptor. In combination with the discovery of the endogenous ligand BigLEN in neuroscience fields and the fact that GPR171 has been used as a T cell marker, the authors used this information to formulate their studies and imply the hypothesis that GPR171 may regulate T cell immunity.

The authors used sophisticated techniques and demonstrate that GPR171 is highly inducible in T cells and serves to suppress T cell activity, proliferation and cytokine release working through the PLC γ 1 ERK/Akt pathway in vitro. They further show that depletion of GPR171 with pharmaceutical antagonists and CRISPR technologies suppresses GPR171 activity which serves to rescue T cell function. Generating hetero and homozygous GPR171 knockout mice, the authors further demonstrate in 2 models (prevention and treatment) of lung melanoma that GPR171 knockout mice have a reduced tumor size and increased survival; all of which hinges on the reversal of GPR171 inhibition of T cell function. The authors finally use their GPR171 antagonists in combination with known immune checkpoint inhibitors to show improved survival of metastatic mice. The results of this study are informative, innovative and, are of interest to an audience beyond the cancer field.

I have a few queries which may require additional experiments to instill confidence in their findings. Please note that these comments below are motivated by the fact that the BigLEN/GPR171 pathway appears to be established. What is not known, and the authors make mention of, is if GPR171 is stimulated by other ligands and if GPR171 works in isolation.

1. What is the concentration of BigLEN and/or proSAAS in the tumor microenvironment? The logic follows that if GPR171 serves to inhibit T cell function in the tumor microenvironment, then BigLEN should be present in sufficient quantities to act on its receptor in order for it to do so. Moreover, are the concentrations of BigLEN used in vitro reflective of the endogenous BigLEN concentration in the tumor microenvironment?

2. GPR171 is highly related to purinergic receptors and, in fact, is close to the gene locus for P2Y receptors. Can the authors confirm that their GPR171 CRISPR target did not affect P2Y receptors, specifically P2Y2, P2Y4 and P2Y12? Along this line, are P2Y receptors intact in GPR171lacZ/lacZ mice?

3. Following from the second query, can the authors confirm that ATP has similar effects in normal and GPR171 targeted cells and animals?

4. The authors reference Colosimo et al. 2019 suggesting that in addition to BigLEN bacterial

derived lipid may stimulate GPR171. Given that the bacterial microbiome is increasingly important in the tumor microenvironment can the authors rule out bacterial stimulation of GPR171?

Minor Comments

1. The authors summarize the information that led them to their question, but do not phrase it in the form of a testable hypothesis. Could the authors provide a hypothesis statement?
2. The authors provide summary data for luciferase experiments but do not appear to provide summary data for calcium imaging. How many cells were imaged and what are the summary results?
3. Image page size is not letter size. This does not conform to NCOMMS formatting.
4. There is no limit to the methods word count in NCOMMS, yet it appears the authors limit the description of a few experimental approaches. Please ensure that methods are divulged explicitly to allow readability, reproducibility and conform to NCOMMS guidelines.

Response to the reviewers' comments

Reviewer #1 (Remarks to the Author):

In the article entitled “The BigLEN/GPR171 pathway suppresses T cell activation and limits antitumor immunity” by Fujiwara and colleagues, the authors describe a novel role for the G protein-coupled receptor GPR171 in regulating T cell-mediated immunity. While a role for BigLEN/GPR171 has been described to regulate food uptake and anxiety, the role for GPR171 in T cells has not been characterized. Here, the authors show that GPR171 acts as an inhibitory receptor on T cells, suppressing T cell receptor signaling, proliferation, and cytokine production. Blocking GPR171 promotes productive anti-tumor T cell responses as a single agent, but has minimal impact on tumor growth, likely because of compensation from other inhibitory receptors. Co-blockade of GPR171 and either PD-1 or TIGIT resulted in enhanced tumor control compared to either single agent treatment, highlighting the potential for GPR171 blockade to augment pre-existing checkpoint blockade modalities. Overall, the authors provide convincing evidence for the role of GPR171 as a negative regulator T cell functions in a variety of different models. The experiments seem well executed and the conclusions are supported by the data provided. There are some points outlined below which need to be addressed prior to publication, which are intended to strengthen an already well executed study.

Thanks for reviewing and providing valuable advises for our manuscript. We have revised our manuscript thoroughly as suggested and have provided additional experimental data and discussions in the revised manuscript.

Major Points:

1.) The most important point that needs to be addressed prior to publication is reporting on the number of independent experiments for each of the results included in the paper. The n within each figure is provided in the Figure Legends or obvious from the way the data is displayed, but the number of independent experiments performed seems to be missing. In order to evaluate the reproducibility of the work, this information needs to be included, preferably in the Figure Legends.

We have included information about the numbers of repeated experiments in the Figure Legend throughout the revised manuscript. During the revision, we have performed additional experiments to make sure that we have replicated all the experiments in the manuscript.

2.) Considering the authors report on a new inhibitory receptor pathway, it would be useful to place this new receptor in context with the other inhibitory receptors. Work from John Wherry's group (Blackburn 2009 PMID 19043418) has shown that in chronic LCMV, many of the inhibitory receptors tend to be co-expressed on the same cells. Can the authors provide information on whether GPR171 also tends to be co-expressed with the other inhibitory receptors, particularly in the tumor setting? There are some data provided on co-expression with Tim-3 (Figure 6A), but inclusion of co-expression with PD-1 and/or TIGIT would also be useful since these are the two inhibitory receptors that the authors follow up on with in vivo blockade.

Thanks for the valuable suggestion. In MC38 TILs from GPR171^{-/-} mice, we co-stained GPR171 (β-gel) with other known immune checkpoints PD1, TIM3, TIGIT, and LAG3. As shown in new Figure 4B of the revision (Page 8), GPR171 is well co-expressed with these

inhibitory receptors in intratumoral T cells, as the reviewer has predicted. In addition, we have cited John Wherry's paper in the revised manuscript.

3.) Can the authors provide a more indepth discussion on the ligand BigLEN, including which cell types produce it, and how this may impact the ability of the BigLEN/GPR171 pathway to regulate T cell functions? Is there any information in the literature/publically available data sets on BigLEN expression outside of the CNS?

Besides the CNS, proSAAS protein (the neuropeptide BigLEN derives from) is found to be expressed by endocrine cells of peripheral tissues (Lanoue E, Endocrinology, 2001). Glycosylated BigLEN has been detected in human and mouse islets by targeted mass spectrometry (Yu Q, Anal Chem., 2017). As a neuropeptide derived from proSAAS, BigLEN could be secreted from those cells or tissues. We did not detect proSAAS expression in our mouse tumor tissues by RT-PCR (Left Figure). Because of the lack of detecting tools, we are still unable to determine the presence of secreted BigLEN in the serum or our tumor models. We have summarized and included this information in the Discussion of the revised manuscript (Page 14).

4.) The authors mention that the BigLEN/GPR171 has mostly been described in the context of regulating feeding and anxiety behaviors. Did the authors observe any impact on these aspects in their studies blocking the GPR171 pathway? A discussion of this aspect of the pathway would be useful to add, especially when considering adding GPR171 blockade to pre-existing immunotherapy regiments. Would issues with feeding/anxiety be a type of “immune-related adverse event” to consider when thinking about immunotherapy in patients. **Thanks for the insightful suggestion. We measured body weight and food intake in our mouse tumor models during GPR171 antagonist treatment for three weeks. GPR171 antagonist didn't affect weight change and food intake in tumor-bearing mice. The results were presented in new Supplementary Figure 5F (Page 11). We also compared body weight and food intake between naive WT and GPR171^{LacZ/LacZ} mice. We did not see any difference of body weight or food intake between WT and GPR171^{LacZ/LacZ} mice at different ages (new Supplemental Figure 4J, Page 7). Therefore, although blockade of GPR171 affects feeding in a short period, GPR171 ablation has no long-term effect on weight gain in mice.**

5.) The authors provide data on GPR171 in both mice and humans, as well as some comparison of expression patterns. Can the authors comment on the amount of homolog between the mouse and human receptors? Is this pathway highly conserved between mice and humans?

GPR171 is highly conserved across human, rat, and mouse, with over 90% protein homology. The 16-aa BigLEN peptide only has three amino acids that are different but similar between mouse and human. We have included the information in Page 13 of the revised manuscript.

6.) The authors provide frequencies of immune cell subsets relative of total in MC38 and B16F10, and there seems to be increased CD8 T cells and decreased Treg cells. Can the authors

provide the CD8/Treg ratios following GPR171 modulation in these models? This is a useful benchmark for thinking about the protective vs. immunosuppressive microenvironments.

We have calculated the CD8/Treg ratio and included in the revision (new Figure 4G, Page 9), which was significantly higher in tumors of GPR171^{LacZ/LacZ} mice.

Minor Points:

1.) In Figure 4G, there should be a label on the data point colored in red, or an explanation of what it is in the Figure Legend. Presumably it is GPR171 lacZ/lacZ without any treatment (or vehicle treated), but this should be indicated in the Figure.

The group colored in red was tumor-bearing GPR171^{LacZ/LacZ} mice without mAb treatment. In the revision, we have added a label to better indicate the group.

2.) The resolution of Figure S1A is very poor, it should be increased to make the labels more legible.

We have replaced it with a figure in high resolution in the revision.

Reviewer #2 (Remarks to the Author):

This article describes a role for GPR171 in T cell activation, with evidence that signaling through GPR171 (e.g. by the neuropeptide BigLEN) has a suppressive influence on signaling through the TCR. These findings are taken further to show that GPR171 inactivation leads to stronger T cell activation, and that this can be beneficial in an anti-tumor response.

Thanks for reviewing our manuscript. In the revision, we have performed additional experiments to address the reviewer' comments and also make sure that we have replicated all the experiments in the manuscript.

1. Overall, this is an exciting and novel finding, of potential translational importance. My major reservation is that there is no evidence that any of these experiments has been performed more than once. For example, for Fig 1, the implication is that they were really only done once: "...PBMCs of a healthy donor...". There are similar issues with all of the data. Indeed, most of the legends don't mention the n for the (technical or biologic) replicates. This is a major issue. Another issue is the statistics, which are mostly not described. For Fig. 4A (left panel) as a - not unique - example: the SEM error bars overlap, yet the p value is given as <0.01. The test is not defined, and frankly, a "significant" p value is a little difficult to believe.

We apologize for the confusion and missing information. We used standard deviation (SD) for error bars of all data and figures, not SEM we mistakenly described. We corrected it and rewrote all figure legends as well as double-checked the statistics. The p-value between dLN and tumor in CD4 T-cell in Figure 4A was 0.0048. We have performed twice or three times of independent experiments and added the information about the numbers of repeats in Figure Legend of the revision.

2. Fig 1 shows data on human T cell expression, where the most expression is found in immune cells with CD8>CD4>everything else. Thymic expression is low. These data are from BIOGPS. Stimulation with aCD3 plus BigLEN reduces signaling by various methods, and the GPR171 gene is expressed in T cells and NK cells in TILs. The signaling reduces Ca²⁺ flux and various cytokine's expression. As noted, it looks like each experiment was only done once. There should also be a control for BigLEN treatment without TCR stimulation. The level of the GPR171 effect on TCR stimulation should be defined more. Is it at the level of CD3, ZAP70 or LAT or later?

We have performed at least twice all these experiments, including Ca²⁺ flux, western blot and cytokine detection. Our Ca²⁺ flux experiment included the BigLEN treatment only group, which was overlapped with the baseline. We have updated the label for figures of Ca²⁺ flux in the revision (Figure 1G).

We also evaluated the effect of BigLEN on proximal TCR signaling, including phosphorylated CD3 ζ and ZAP70. BigLEN is able to clearly inhibit the phosphorylation of both CD3 ζ and ZAP70. These results were included in Figure 1J and Supplementary Figure 2E.

3. The BIOGPS data for GPR171 expression of in mouse T cells are not shown. Strangely, they are very different to the human data (<http://biogps.org/#goto=genereport&id=229323>), with mast cells>DC>NK>B>T, and rather low expression in thymus. This discrepancy is not mentioned. The PCR data (suppl 3A) are not quantitative and therefore not very useful.

The expression of mouse GPR171 seems to be broader than human GPR171. A previous publication examining the expression of mouse GPR171 supported that, though mouse GPR171 is primarily expressed in lymphoid lineages, but not myeloid cells (Experimental Hematology, 2013). We have included the BIOGPS data for mouse GPR171 in new Supplemental Figure 3A. We also performed quantitative PCR to carefully determine GPR171 expression in immune cells of naïve mouse spleen. We did find substantial GPR171 transcript in mouse B cells, which is very different from human GPR171 (new Supplemental Figure 3C). However, we did not find any detectable GPR171 expression by beta-gal staining in naïve mouse splenocytes (Supplemental Figure 4G), suggesting that there is a tight control for GPR171 translation.

This said, the data on inhibition of OT-I CD8+ cell and CD4+ cell stimulation look good (how many times were they done?).

All the OT-1 and CD4+ T cell assays in Figure 2 have been done twice. We have listed the numbers of repeats in all experiments in the Figure Legend of the revised manuscript.

4. Using the antagonist of GPR171, MS0021570, they find that the T cells respond better to TCR stimulation, and there is more expansion in tumor-bearing mice.

Thanks for the encouraging comment. We hope the GPR171 blockade can become one option for cancer immunotherapy.

5. The authors made a knockin mouse for the GPR171 locus that replaced the coding part of GPR171 with LacZ. GPR171^{+/LacZ} mice show lots of expression of LacZ in the thymus, more in the early DN thymocytes than in the DP or SP subsets. BIOGPS did not show this (DN not shown, DP lower than SP), so I think there should be more dissection of the expression in normal mice.

As the reviewer suggested, we sorted out each thymocyte subset from WT B6 mice to carefully quantify GPR171 expression by qPCR in Supplementary Figure 4E. We further re-examined the expression of GPR171 in thymocytes from GPR171^{+/LacZ} mice by flow cytometry of β-gal. The results did show the highest GPR171 expression in SP thymocyte, followed by DP, then the lowest in DN, which is consistent with the BIOGPS data. These results are now presented in Figure 3A and Supplementary Figure 4E.

6. The “knockout” mice (LacZ/LacZ) responded better than LacZ/+ mice to TCR stimuli, and LacZ/+ responded worse than WT when BigLEN was added to stimulate the GPR171. So, GPR171 allows suppression of TCR signals by BigLEN. In antitumor responses, the LacZ/LacZ “knockout” worked better, and depletion of CD4, CD8 or NK cells all reduced the effect of higher activity in the LacZ/LacZ mice.

BigLEN inhibited TCR-stimulation by CD3 mAb via GPR171 *in vitro* and ablation of GPR171 in mice leads to improved antitumor immunity. The improved antitumor effect in GPR171^{-/-} mice appears to require CD4+ T cells, CD8+ T cells, as well as NK cells.

7. The antagonist MS0021570 allowed more TCR signaling, resulting in higher expression of checkpoint inhibitory molecules PD-1 and TIGIT. aPD-1 caused upregulation of GPR171 on TILs. There are intriguing data showing that MS0021570 works synergistically with different

checkpoint blockade antibodies (best was a combination of aPD-L1 + aCDTA4).

The data are good and interesting, but they do not appear to have been replicated.

Thank you for reviewing our data. Throughout the revised manuscript, we have included the numbers of repeated experiments in the Figure Legends.

Minor points:

1. OT-I, not OT1.

BALB/c, not Blab/c or Balb/c.

We have corrected the words in our revised manuscript.

2. Repetition in Discussion

1st para:

“GPR171 expression is inducible in both human and mouse T cells, and amplification of GPR171 signaling via BigLEN delivers a suppressive signal to T cells.”

2nd para:

“GPR171 is inducible in T cells of both human and mouse, and GPR171 engagement by BigLEN inhibits TCR-mediated signaling cascades.”

We have removed the description in 1st paragraph to avoid the repetition.

3. Suppl Fig 3D scale needs to be “broken” or changed to log, since most of the cytokines are expressed too weakly to see the differences (apparently very statistically sig) on the graph.

As the reviewer suggested, we have broken down individual cytokine in the revised Supplement Figure 3F to better reveal the differences.

Reviewer #3 (Remarks to the Author):

Fujiwara et al studied how GPR171 regulates T cell function in relation to T-cell immune checkpoint activity. The authors used their in vitro results to inform specific in vivo experiments in order to study the role of GPR171 in the tumor microenvironment and how it may act as a novel immune checkpoint. The GPR171 is a GPCR which, until recently was classified as an orphan receptor. In combination with the discovery of the endogenous ligand BigLEN in neuroscience fields and the fact that GPR171 has been used as a T cell marker, the authors used this information to formulate their studies and imply the hypothesis that GPR171 may regulate T cell immunity.

The authors used sophisticated techniques and demonstrate that GPR171 is highly inducible in T cells and serves to suppress T cell activity, proliferation and cytokine release working through the PLC γ 1 ERK/Akt pathway in vitro. They further show that depletion of GPR171 with pharmaceutical antagonists and CRISPR technologies suppresses GPR171 activity which serves to rescue T cell function. Generating hetero and homozygous GPR171 knockout mice, the authors further demonstrate in 2 models (prevention and treatment) of lung melanoma that GPR171 knockout mice have a reduced tumor size and increased survival; all of which hinges on the reversal of GPR171 inhibition of T cell function. The authors finally use their GPR171 antagonists in combination with known immune checkpoint inhibitors to show improved survival of metastatic mice. The results of this study are informative, innovative and, are of interest to an audience beyond the cancer field.

I have a few queries which may require additional experiments to instill confidence in their findings. Please note that these comments below are motivated by the fact that the BigLEN/GPR171 pathway appears to be established. What is not known, and the authors make mention of, is if GPR171 is stimulated by other ligands and if GPR171 works in isolation.

We appreciate the reviewer's positive comments on our manuscript and believe that your recommendation could help our projects.

1. What is the concentration of BigLEN and/or proSAAS in the tumor microenvironment? The logic follows that if GPR171 serves to inhibit T cell function in the tumor microenvironment, then BigLEN should be present in sufficient quantities to act on its receptor in order for it to do so. Moreover, are the concentrations of BigLEN used in vitro reflective of the endogenous BigLEN concentration in the tumor microenvironment?

Because of the lack of detecting tools, we are still unable to determine the presence of secreted BigLEN in the serum or our tumor models. Otherwise, we evaluated the expression of PCSK1N expression in the various types of tumors (B16-OVA, MC38 and CT26), and mouse brain tissue was used as a positive control. The expression of PCSK1N in the tumors was undetectable (left figure). Therefore, questions remain whether there is soluble BigLEN in the tumors, or there is additional ligand for GPR171 which limits antitumor immunity in our tumor models.

2. GPR171 is highly related to purinergic receptors and, in fact, is close to the gene locus for P2Y receptors. Can the authors confirm that their GPR171 CRISPR target did not affect P2Y receptors, specifically P2Y2, P2Y4 and P2Y12? Along this line, are P2Y receptors intact in GPR171lacZ/lacZ mice?

We did qPCR to quantify the expressions of P2Y receptors between Jurkat^{WT} and Jurkat^{KO} cell as well as T cells isolated from GPR171^{+/+} or GPR171^{LacZ/LacZ} mice. There is no difference of expression in P2Y receptors we examined.

3. Following from the second query, can the authors confirm that ATP has similar effects in normal and GPR171 targeted cells and animals?

As the reviewer suggested, we evaluated the effects of Ca²⁺ influx by ATP, which was purchased from Sigma-Aldrich (A2383). ATP at 500μM or 1mM significantly activated Ca²⁺ influx in CHO cells but didn't trigger calcium flux in Jurkat cells, naïve splenocytes, or activated OT-1 splenocytes by OVA peptide (as indicated below). Therefore, we were unable to evaluate the effect of Ca²⁺ influx by ATP alone on Jurkat or mouse T cells (both naïve and activated). Nevertheless, because our qPCR data indicated that our GPR171^{-/-} Jurkat cells or T cells from GPR171^{-/-} mice have similar levels of P2Y receptors, GPR171 ablation itself should not affect the effect of ATP.

4. The authors reference Colosimo et al. 2019 suggesting that in addition to BigLEN bacterial derived lipid may stimulate GPR171. Given that the bacterial microbiome is increasingly important in the tumor microenvironment can the authors rule out bacterial stimulation of GPR171?

To evaluate the effect of GPR171 ligands derived from microbiome in the gut, B6 mice were treated by antibiotics water which includes amoxicillin (1mg/ml), streptomycin (5mg/ml) and colistin (1mg/ml) for two weeks and then inoculated MC38 subcutaneously. The GPR171 antagonist treatment or vehicle were treated soon after MC38 inoculation. The results were shown in Supplementary Figure 6, Page 14.

Minor Comments

1. The authors summarize the information that led them to their question, but do not phrase it in the form of a testable hypothesis. Could the authors provide a hypothesis statement?

We have rewritten our manuscript to present our study in a more hypothesis-driven way.

2. The authors provide summary data for luciferase experiments but do not appear to provide summary data for calcium imaging. How many cells were imaged and what are the summary results?

As the reviewer suggested, we have provided the raw data of Ca²⁺ flux in Source Data file. We evaluated 700 cells per second in each sample. In addition, we calculated area of the curve (AUC) and compared in each group as a summary result. These results were updated in the revised manuscript (Figure 1G and supplementary figure 2B).

3. Image page size is not letter size. This does not conform to NCOMMS formatting.

We have changed image sizes to fit NCOMMS guidelines.

4. There is no limit to the methods word count in NCOMMS, yet it appears the authors limit the description of a few experimental approaches. Please ensure that methods are divulged explicitly to allow readability, reproducibility and conform to NCOMMS guidelines.

As the reviewer suggested, we rewrote the section of material and methods to include more experimental details.

REVIEWERS' COMMENTS

Reviewer #1 (Remarks to the Author):

In the original submission of the article entitled "The BigLEN/GPR171 pathway suppresses T cell activation and limits antitumor immunity," Fujiwara and colleagues described a novel role for GPR171 in regulating anti-tumor T cell responses. The revised manuscript includes a number of new discussion points, new data about how GPR171 expression relates to the other inhibitory receptors and alterations in the CD8/Treg ratio in the in vivo experiments, and additional details on statistical tests performed and the number of independent experiments performed. These additions have strengthened the manuscript, and the revised version adequately addresses all of my points.

Reviewer #2 (Remarks to the Author):

My major issue with the original version of the paper was lack of evidence of reproducibility. The authors now provide this information. Most experiments have been repeated 2x and others 3x. They also provide data and discussion of the different expression patterns of the human and mouse versions of GP171.

Reviewer #3 (Remarks to the Author):

Fujiwara et al. have produced a good faith effort and responded to most of this reviewers comments. This reviewer appreciates the additional data produced to address the comments raised by all reviewers. Also, this reviewer appreciates the care the authors have taken to revise the manuscript and reformat it as per the journal's guidelines.

I only have one remaining comment to make. In response to my initial comment regarding the quantification of ProSAAS and/or BigLEN in the tumour microenvironment or serum, the authors responded by stating that there is a lack of methods to quantify this peptide and its precursor. They provide data to demonstrate that the positive control PCSK1N is not expressed. However, these data do not completely answer my query. Stating there are lack of quantitative tools is partially true. LC/MS has been used to quantify both ProSAAS and BigLEN in the brain (PMID: 28864778) and cultured neuroendocrine cells (PMID: 11742530). The reviewer agrees that there are likely other ligands for GPR171 but, such a search is far beyond the scope of this paper. Further comment on the quantification of these ligands may provide readers with more information regarding the relationship of this peptide, its receptor and T cell immune checkpoint activity.

Reviewer #1 (Remarks to the Author):

In the original submission of the article entitled “The BigLEN/GPR171 pathway suppresses T cell activation and limits antitumor immunity,” Fujiwara and colleagues described a novel role for GPR171 in regulating anti-tumor T cell responses. The revised manuscript includes a number of new discussion points, new data about how GPR171 expression relates to the other inhibitory receptors and alterations in the CD8/Treg ratio in the in vivo experiments, and additional details on statistical tests performed and the number of independent experiments performed. These additions have strengthened the manuscript, and the revised version adequately addresses all of my points.

We thank the reviewer for constructive suggestions and valuable advices.

Reviewer #2 (Remarks to the Author):

My major issue with the original version of the paper was lack of evidence of reproducibility. The authors now provide this information. Most experiments have been repeated 2x and others 3x. They also provide data and discussion of the different expression patterns of the human and mouse versions of GPR171.

We thank the reviewer for constructive advices.

Reviewer #3 (Remarks to the Author):

Fujiwara et al. have produced a good faith effort and responded to most of this reviewers comments. This reviewer appreciates the additional data produced to address the comments raised by all reviewers. Also, this reviewer appreciates the care the authors have taken to revise the manuscript and reformat it as per the journal's guidelines.

I only have one remaining comment to make. In response to my initial comment regarding the quantification of ProSAAS and/or BigLEN in the tumour microenvironment or serum, the authors responded by stating that there is a lack of methods to quantify this peptide and its precursor. They provide data to demonstrate that the positive control PCSK1N is not expressed. However, these data do not completely answer my query. Stating there are lack of quantitative tools is partially true. LC/MS has been used to quantify both ProSAAS and BigLEN in the brain (PMID: 28864778) and cultured neuroendocrine cells (PMID: 11742530). The reviewer agrees that there are likely other ligands for GPR171 but, such a search is far beyond the scope of this paper. Further comment on the quantification of these ligands may provide readers with more information regarding the relationship of this peptide, its receptor and T cell immune checkpoint activity.

We thank the reviewer for providing constructive suggestions. Accordingly, we have added more discussions about detecting methods for BigLEN and other potential ligands for GPR171.